# Application of Induced Pluripotent Stem Cells (iPSCs) in Hereditary and Viral Diseases of the Liver: Modeling and Treatment

**DOI:** 10.3390/ijms26199432

**Published:** 2025-09-26

**Authors:** Vladimir Andriianov, Alina Malyutina, Egor Panferov, Alexander Karabelsky, Roman Ivanov, Ekaterina Minskaia, Vasiliy Reshetnikov

**Affiliations:** Translational Medicine Research Center, Sirius University of Science and Technology, 354340 Sochi, Russia; andriyanov.vs@talantiuspeh.ru (V.A.);

**Keywords:** induced pluripotent stem cells, iPSCs, liver disease, disease modeling, liver disease modeling, liver disease treatment

## Abstract

The high prevalence and diversity of liver diseases present a significant problem for modern healthcare. Despite FDA approval of gene therapy drugs to treat hemophilia A and B, available treatment methods for other hereditary liver diseases are mainly limited to the frequently ineffective traditional therapies and surgical intervention. In recent years, significant progress has been made in the treatment of hepatitis C, but hepatitis B is still considered an incurable disease. In this regard, the treatment of hereditary and viral liver diseases using gene or cell therapy remains relevant. This review is focused on the current state of the induced pluripotent stem cells (iPSCs) field in the context of modeling and treatment of hereditary, viral, and some other liver diseases, both ex vivo and in vivo. Here we present a detailed discussion of the possible ways of modeling liver diseases ex vivo using iPSCs (reprogramming of patient somatic cells and genetic engineering (GE) of healthy iPSCs), summarize gene editing (GE) and non-GE approaches for the treatment of liver diseases, and demonstrate that iPSCs and their derivatives are widely used to treat liver diseases in vivo. Taken together, we are presenting a comprehensive analysis of 2D and 3D iPSC-based products in the context of liver diseases, discussing the advantages and disadvantages of this platform, including the comparison with other types of stem cells and animal models. This analysis may help understand not only the potential but also the limitations associated with the use of iPSCs in the context of various types of liver diseases.

## 1. Introduction

Liver disease has become one of the most significant global health problems in recent decades. According to the World Health Organization, the number of deaths associated with hepatitis continues to rise [1]. Of particular concern is the increasing incidence of non-alcoholic fatty liver disease, which is estimated to affect approximately 32.4% of the world’s population and continues to grow [2]. There is also an increase in the incidence of alcoholic liver disease, especially among young people and women.

Today, various approaches are used for the treatment of liver diseases, such as drug therapy and surgical intervention (including liver transplantation). For example, direct-acting antiviral drugs are used for the treatment of viral hepatitis C with a 95% success rate [3]. In certain cases where traditional treatment is ineffective, liver transplantation, one of the most radical and difficult-to-access treatment methods, can be used [4]. These data highlight the need to develop new strategies to combat liver diseases, including cell therapy.

In recent years, increasing attention has been given to the use of induced pluripotent stem cells for the treatment of liver diseases. iPSCs have a unique ability to differentiate into various cell types, making them a promising tool for regenerative medicine. Soon after the development of iPSC technology, it became clear that these cells could be used to model diseases, screen drugs, and develop personalized treatments [5]. For example, iPSCs are used to test the toxicity of drugs such as thalidomide [6] and model diseases such as amyotrophic lateral sclerosis [7].

It is known that the origin, production protocols, and maintenance of iPSCs can influence the reprogramming efficiency and speed, as well as the properties of the resulting cells [8], which leads to the requirement of careful control of iPSC production and characterization. Existing clinical trials mainly use off-the-shelf human leukocyte antigen-matched donor iPSCs instead of autologous iPSCs, which allows for a significant reduction in production time and costs [9]. Such donor iPSCs are stored in specialized cell banks, the number of which is growing along with the increasing demand for iPSC-based products [10]. The emergence and expansion of iPSC banks in different countries around the world significantly simplifies these tasks, as it ensures the standardization of reagents and methods. By 2019, there were at least 10 iPSC banks in the world [11]. This diversity opens up new opportunities for basic and translational research. Thus, the iPSC platform represents a promising direction in the treatment of liver diseases, although its widespread use still requires further research and clinical trials.

Currently, already several dozen iPSC-based products are at various stages of clinical trials [12,13]. iPSCs are used to treat a variety of diseases, including retinopathies, cardiovascular and neurodegenerative diseases, and cancer. With the help of iPSCs, scientists create liver disease models to understand the underlying mechanisms driving the development of pathologies [14,15,16,17,18] and to develop new effective treatment approaches. A number of reviews have been published on the iPSC platform in recent years [19,20,21,22,23,24,25,26,27,28]. Previously published reviews on this subject often focused on a limited number of hereditary liver diseases [19,21,22,23,24,26,27], either focusing on other liver diseases or discussing a wide list of hereditary disorders while only touching upon a limited number of studies devoted to modeling using iPSCs [20,25,28] and giving a description of in vivo studies. The aim of the present study, however, is to provide a comprehensive literature review on modeling and treatment of hereditary and viral liver diseases using iPSCs. We analyzed the studies that examined various inherited and viral liver diseases; however, this review did not include studies that examined diseases such as NAFLD, NASH, and drug-induced liver disease. To the best of our knowledge, this review greatly contributes to the field of iPSC-based treatment of monogenic and viral liver diseases at the level of pre-clinical studies. Main sources of iPSCs, iPSC-derived cells, and their application in the context of modeling and studying liver diseases discussed in this review are demonstrated in Figure 1.

## 2. Classification and Statistics

In this review, the following classification of hereditary liver diseases proposed earlier is used [29]: monogenic diseases with primary hepatic expression without significant parenchymal (parenchyma—functional tissue of the liver) damage; monogenic diseases with primary hepatic expression and parenchymal damage; and monogenic diseases with both hepatic and extrahepatic expression. Viral liver diseases and a group of diseases that could not be classified into the groups described above are also highlighted.

Hemophilia A (HA) and B (HB), urea cycle disorders (UCD, ornithine transcarbamylase deficiency (OTCD), carbamoyl phosphate synthetase 1 deficiency (CPS I deficiency), citrullinemia type 1 (CTLN1), and arginase deficiency (AD)), hypercholesterolemia (autosomal recessive and familial), primary hyperoxaluria type 1 (PH1), wild-type transthyretin amyloid (WTTA), Crigler-Najjar syndrome, familial amyloid polyneuropathy (FAP) (transthyretin (TTR)-related amyloidosis), and glycogen storage disease type 1a are monogenic disorders with primary hepatic expression without significant parenchymal damage that have been studied via the iPSC platform.

iPSCs have also been used for monogenic disorders with primary hepatic expression and parenchymal damage: alpha-1 antitrypsin deficiency (AATD), Alagille syndrome (ALGS), progressive familial intrahepatic cholestasis (PFIC), argininosuccinate lyase deficiency (argininosuccinic aciduria, ASA, ASL), Wilson’s disease (WD), MEDNIK syndrome, glycogen storage disease type Ib (GSDIb), and tyrosinemia 1.

The results of iPSC-based studies for monogenic diseases with both hepatic and extrahepatic expression—mitochondrial DNA depletion syndrome (MDS), Alpers-Huttenlocher syndrome (AHS), Niemann–Pick type C (NP-C), Wolman disease, Pompe disease (glycogen storage disease type II), cystic fibrosis (CF), and abetalipoproteinemia (ABL)—are also presented.

The iPSC platform has been used for viral liver diseases, such as hepatitis B, C, and E, and severe acute respiratory syndrome-related coronavirus 2 (SARS-CoV-2). And finally, we identify a group of other liver diseases that are neither monogenic nor viral but are related to the liver and are discussed in works devoted to iPSCs. Among them are biliary atresia (BA), tetralogy of Fallot (TOF), and polycystic liver disease (PLD).

Table 1 shows various types of hereditary, viral, and other liver diseases discussed in this review and also describes disease prevalence and phenotype.

Overall, the results of more than 100 experimental studies devoted to the use of iPSCs in the context of liver disease modeling or therapy are examined in this review.

## 3. Ex Vivo Studies

The development of new approaches towards disease modeling is an important aspect of modern experimental biology, owing to its great importance both in basic and applied biomedical research. Existing liver models can be categorized into two major groups: ex vivo and in vivo models.

Ex vivo models derived from human cells are more relevant for modeling human diseases compared to laboratory animals due to better predictability and the lack of interspecies differences [58] while being more scalable, making them especially suitable for high-throughput studies, for example, deciphering the action mechanism of novel drugs or unraveling the molecular basis of liver diseases [59]. Furthermore, cell-based assays also reduce animal use in accordance with the 3Rs principle [60].

Ex vivo models can be broadly divided further into two groups: 2D and 3D cell cultures. Two-dimensional models include the primary hepatocyte (PHH) cultures [61], immortalized liver cells such as HepG2 cells [62], and iHLCs [63]. These cells differ in physiological relevance, ease of acquisition, and their survivability in long-term experiments, as summarized in Table 2 [64,65].

A major disadvantage of 2D models is their inability to represent the full complexity of tissues and organs. The lack of 3D interactions both between different cell types and between cells and the extracellular matrix limits their ability to accurately simulate the true physiological state of the tissue, necessitating the use of more complex 3D models [58]. Such models include spheroid and organoid liver cultures [79], organ-on-a-chip platforms [80], and the slice cultures [81].

iHLCs are particularly promising as liver models, sharing enough similarities with the primary hepatocytes to be significantly more clinically relevant than cancer cell lines, while having a much higher (albeit still limited) throughput compared to the primary hepatocyte cultures. Different protocols for deriving iHLCs from iPSCs exist; however, they all share the trait of including the definitive endoderm (DE) and hepatic progenitors as intermediate stages before the hepatocyte differentiation [22]. Many variations of the differentiation protocols exist. DE induction is typically achieved by the Activin A growth factor, while the hepatic specification and maturation require the fibroblast (FGF) and hepatocyte (HGF) growth factors [82]. Both on-feeder and feeder-free iPSC cultures have previously been successfully differentiated into this transitory endoderm stage, with the latter option often being preferable due to the reduced risks of contamination and the overall reduced fluctuations in differentiation efficiency associated with it [83]. Hepatocyte differentiation, likewise, can be achieved in multiple ways, with HGF, oncostatin M, and dexamethasone being some of the most common reagents, with epidermal growth factor (EGF), FGF4, insulin, hydrocortisone, and other cytokines or their small molecule surrogates sometimes being used as well.

This stepwise differentiation protocol results in cells expressing adequate levels of albumin, cytokeratin 18 (CK18), cytochrome p450 isoforms, alpha-1 antitrypsin (AAT), and asialoglycoprotein receptor 1 [84]. It is important to note that compared to primary hepatocyte cultures, genes related to the bile secretion pathway (notably the *AQP9* and *UGT1A1*), the tricarboxylic acid (TCA) cycle, and drug metabolism (most prominently the cytochrome P450 (CYP) and cytochrome P450 2E (CYP2E)) are significantly downregulated in iHLCs [77]. Furthermore, regardless of the protocol used, the resulting iHLCs invariably exhibit a fetal-like phenotype, characterized by a reduced activity of key enzymes (as low as ~0.1% of that in primary hepatocytes), such as the mature isoforms of cytochrome p450, namely the CYP2A6 and CYP3A4 detoxification enzymes, as well as a high level of alpha fetoprotein expression [76]. Nonetheless, such cells are considered to be capable of reproducing key features of inherited, acquired, and infectious diseases to a reasonable degree of accuracy [85].

Compared to 2D, 3D cultures, and particularly the organoid cultures, while significantly more challenging and expensive to produce and maintain, they offer better representation of in vivo conditions at the cost of reduced reproducibility [86]. Organoids, as defined by Hans Clevers [87], are “a 3D structure grown from stem cells and consisting of organ-specific cell types that self-organizes through cell sorting and spatially restricted lineage commitment.” iPSC-derived hepatic organoids expand upon the conventional 2D differentiation protocols with additional steps, typically by culturing iPSC-derived hepatoblasts on Matrigel or other extracellular matrices to achieve 3D growth. Other means, such as iPSC-derived endoderm cells’ self-assembly and organization in a rotating wall vessel without Matrigel [88] or co-culturing iPSC-derived hepatoblasts with mesenchymal stem cells (MSCs) and endothelial cells, have also been reported [89]. Compared to the 2D, the 3D organoid cultures offer the arrangement of correctly polarized different constituent liver cells, allowing for its physiological functions, particularly the metabolic and secretory, to be recapitulated correctly, while also supporting the hepatocytes in a differentiated state. Many types of these organoid cultures exist, from the comparatively simple aggregates and condensates to the elaborate hepatobiliary organoids and organoids with stromal inclusions, and have been extensively reviewed elsewhere [90].

A total of eighty-two studies dedicated to the ex vivo modeling (both with and without treatment) of liver diseases were analyzed. The strategies for ex vivo modeling of various liver diseases based on iPSCs are shown in Figure 2. Typically, modeling is achieved by reprogramming the somatic cells of the patients into iPSCs. However, three studies included both the reprogramming and editing of healthy cells (mutagenesis). iHLCs derived from healthy iPSCs were also used to model viral infections in 16 studies: hepatitis (15 studies) and SARS-CoV-2 (1 study). Interestingly, while the modeling of viral infections relied on the viral infection of healthy cells, hereditary diseases were typically modeled by reprogramming somatic donor cells with the necessary genotype.

All ex vivo treatment studies can be divided into 3 groups depending on the type of therapy used: 28 studies used non-gene engineering methods (small molecules, peptides, adenoviruses, small interfering RNA (siRNA), and antisense oligonucleotides), 22 studies used gene engineering (CRISPR/Cas9), transcription activator-like effector nucleases (TALEN), and lentiviruses), and finally, 5 studies combined both approaches. Perhaps unsurprisingly, gene engineering approaches were exclusively used for the treatment of hereditary diseases (22/22), while antiviral treatment strategies conversely utilized exclusively non-gene engineering approaches (11/11). A detailed description of all studies, including, among others, the types of iPSC-derived cells and treatment methods (for those studies where therapy was used), as well as the summary and key findings, is presented in Appendix A.

### 3.1. Monogenic Diseases with Primary Hepatic Expression Without Significant Parenchymal Damage

When urea cycle disorders occur, citrullinemia type 1 is an autosomal recessive disorder caused by mutations in the gene *ASS1*, encoding the argininosuccinate synthase (ASS), an enzyme responsible for the third step of the urea cycle. Differences in ammonia metabolism between animals and humans make human iHLCs a particularly promising choice for screening the potential disease treatment strategies, given that the ASS is predominantly produced in periportal hepatocytes. For example, ammonia clearance in iHLCs differentiated from PBMC-derived iPSCs of a patient with CTLN1 was shown by Yoshitoshi-Uebayashi et al. [92] to be improved with the addition of L-arginine. This approach also significantly increased the concentration of urea, suggesting a partial restoration of the functional activity of the urea cycle by this amino acid. The authors also demonstrated that ASS deficiency has an indirect impact on metabolic processes other than the urea cycle. Many components of the TCA cycle were found to be present in higher quantities in CTLN1 cells than in the control group, likely due to a diversion of the metabolic flux from the urea cycle into the TCA cycle, as the two are interconnected through both aspartate and fumarate. Another potential treatment strategy, successfully demonstrated by Akbari et al. [93] in CTLN1 hepatic organoids, is the overexpression of the wild-type *ASS1*. This too resulted in the reduction in ammonia concentration in culture media and the partial restoration of ureagenesis, confirming a partial rescue of the phenotype and showing both this particular model and the disease in general to be amenable to genetic manipulation.

Another disease that similarly affects the urea cycle in the liver is Arginase-1 deficiency, which is caused by the mutations in the *ARG1* gene. These mutations were shown to be largely correctable by CRISPR/Cas9 gene editing. Lee et al. [94] demonstrated that the insertion of a codon-optimized *ARG1* expression cassette into Exon 1 of the endogenous hypoxanthine-guanine phosphoribosyltransferase locus in hiPSCs derived from hyperargininemic patients results in significantly elevated expression levels of Arginase-1 compared to uncorrected hiPSCs, which persists after differentiation into iHLCs. A similar approach by Sin et al. [95] involved the incorporation of edited exons 7 and 9 (which were previously excised to model the AD) into the intron 6 of the *ARG1* locus in murine iPSCs, using the *piggyBac* transposon. Successful reintroduction of the exons using CRISPR/Cas9 homology-directed repair (HDR) and subsequent hepatic differentiation resulted in a higher level of urea production in repaired iHLCs; however, it was still appreciably lower than in wild-type hepatocytes, which the authors presume might be due to the insufficient maturity of obtained iHLCs.

Ornithine transcarbamylase deficiency is also a urea cycle disorder and an X-linked inherited disease. The only known function of ornithine transcarbamylase (OTC) is catalyzing the formation of citrulline from ornithine and carbamylphosphate, either during the urea cycle or during arginine biosynthesis. Consequently, OTC loss of function results in severe hyperammonemia. iPSC lines from OTCD patients have been successfully generated, with both nucleotide substitutions (c.663 + 2T > G, slipping) [96] and exon deletions (exons 3–9) being reported [97]. The first known use of such OTCD patient-derived iPSCs as a model for treatment efficiency by Zabulica et al. [98] involved the use of CRISPR/Cas9 HDR to repair the disease-causing mutations in the *OTC* gene, with no off-target editing seemingly occurring. The resulting iHLCs and hepatic organoids exhibited a gradual increase in urea production, suggesting a partial restoration of the urea cycle following the gene correction. It should be noted, however, that the obtained results were still significantly less pronounced compared to healthy primary hepatocytes, owing to the phenotype limitations of the iHLCs. One of the potential reasons for this discrepancy was identified as the lack of expression of Aquaporin 9 (AQP9) in iHLCs, a trait they share with the fetal liver cells. In a study by Laemmle et al. [99], AQP9 was overexpressed in iHLCs, which was shown to normalize the urea secretion in iHLCs, potentially opening the way for more clinically relevant iPSC disease modeling of urea cycle disorders in the future.

Familial hypercholesterolemia (FH) is characterized by highly elevated levels of low-density lipid-cholesterol (LDL-C) in blood and is primarily (in 80% of monogenic cases) caused by a mutation in the low-density lipoprotein receptor (LDLR). Other important mutations include the defective apolipoprotein B-100 (Apo-B100), an important component of LDL, and a gain-of-function mutation in proprotein convertase subtilisin/kexin 9 (PCSK9). FH is primarily an autosomal dominant disorder, although rare cases of autosomal recessive hypercholesterolemia (ARH) forms have been described as well [100]. iHLCs generated from FH patient-derived iPSCs are known to successfully recapitulate FH pathophysiology in culture, most notably the deficient LDL-C uptake, lack of response to lovastatin, and an elevated secretion of ApoB-100, making them a useful model for studying aberrant lipid and cholesterol metabolism [101]. ARH iHLCs have also been shown to recapitulate the LDL uptake and the functional defects of the LDLRAP1 mutations characteristic of the disease reasonably well [14]. This remarkably accurate representation of the pathogenic phenotype allows for the screening of LDL-C-reducing pharmaceutical compounds with varying therapeutic mechanisms, crucially including the LDLR-independent ones: since most common LCL-C-reducing drugs act through modulating the LDLR, which is nonfunctional in FH, they consequently have little effect for FH treatment. In particular, as shown in a study by Cayo et al. [102], cardiac glycosides, which are normally used to treat heart failure, were shown to be able to reduce both the cellular and secretory ApoB-100 levels of iHLCs, seemingly through a post-translational enhancement of ApoB-100 turnover. CRISPR/Cas9 gene editing is another promising approach towards potential FH treatment. For example, Omer et al. [103] demonstrated that repairing the 3-base deletion in Exon 4 of the *LDLR* gene of patient-derived iPSCs allowed for the iHLCs to be obtained after subsequent differentiation to regain normal levels of LDLR-mediated LCL endocytosis. Similarly, in a study by Fattahi et al. [104], genetic transformation of FH-iHLCs with a lentiviral vector carrying the functional variant of the *LDLR* gene has been shown to be capable of normalizing LDLR function and restoring the LDL uptake by the cells as well.

Hemophilia A is caused by a deficiency in coagulation factor VIII (FVIII) clotting activity [105], while hemophilia B is caused by a deficiency in factor IX (FIX) [106]. Both factors are primarily synthesized in the liver; however, whilst FIX is predominantly synthesized by hepatocytes, FVIII is primarily produced by sinusoidal endothelial cells, making different models necessary for disease modeling. Both hemophilia A and hemophilia B iPSC lines are well established, exhibit satisfactory hepatic differentiation, and produce mutated forms of their respective coagulation factors with suitably dampened activity [107,108].

A particularly important aspect of hemophilia A modeling is the testing of gene therapy approaches, which this disease is remarkably responsive to; even a small increase in secreted FVIII levels results in a significant improvement of symptoms. This can be achieved, for example, by CRISPR/Cas9-mediated target-specific knock-ins of B-domain-deleted forms of *FVIII* cDNA (beneficial due to their compact size, resulting from the deletion of exon 14, which facilitates both the gene delivery itself and the subsequent expression) into the *FVIII* locus, resulting in iPSC lines suitable for downstream autologous cell therapy [109]. Due to a large number of unique mutations and difficulties in targeting the endogenous mutation site, this approach is not always feasible. Park et al. [110] demonstrated that the insertion of the corrected gene into a safe harbor site instead (such as the *H11* locus) could result in a universal correction regardless of the individual mutation, using the same subgenomic RNA (sgRNA) and donor DNA. Another gene editing tool used for gene correction of FVIII is TALENs, which are particularly useful for reversing inversions. Nearly 50% of severe hemophilia A cases are caused by inversions in intron 22, and 1–4% are caused by inversions in intron 1. TALEN pairs are useful both for inducing these reversions in iPSC systems for modeling purposes and for their correction by reversion, with the same TALEN pair being able to be used in both cases, as demonstrated by Park et al. [15] for intron 1. Similarly, the first report by Wu et al. [63] about the large targeted gene insertion used to replace an inversion in *FVIII* intron 22 involved the use of TALENs targeting the junction between exon 22 and intron 22, rescuing both the transcription and secretion of FVIII in gene-corrected iPSC-derived endothelial cells.

On the other hand, since FIX is predominantly synthesized by hepatocytes, hemophilia B is typically modeled using iHLCs. Gene therapy approaches are likewise commonly used, with a particular mutation of the *FIX* gene, g.31134 G > T, p.R384L, being especially important for therapeutic purposes, since it results in functional FIX with heightened activity, called FIX-Padua. Luce et al. [111] compared the efficiency of FIX-Padua synthesis in patient-derived 2D and 3D organoid liver cultures. The FIX minigene was inserted into the AAVS1 site (AAV integration site 1 within the first intron of the phosphatase 1 regulatory subunit 12C gene on chromosome 19) under the hepatic-specific apolipoprotein A2 promoter. As a result, the FIX produced in iHLCs was functionally impaired and lacked certain post-translational modifications, most notably the γ-carboxylation of the gamma-carboxyglutamic acid-rich domain of the protein, necessary for the FIX to assume the correct conformation crucial for its activity. iHLCs are incapable of performing these modifications due to their immature phenotype, lacking the necessary enzymes. Organoid cultures, on the other hand, produced the complete form of FIX. Interestingly, few other papers mention this complication with the in vitro evaluation of FIX activity, perhaps highlighting the importance of the exact differentiation protocols and gene editing strategies chosen for the study. For example, Tang et al. [112] seemingly encountered no such issues when testing the activity of FIX-Padua secreted from gene-corrected iHLCs. In this study, full-length FIX-Padua was inserted before the *FIX* initiation codon into patient-derived iPSCs using CRISPR/Cas9. The resulting iHLCs, differentiated from site-specific integrated clones, secreted modified FIX with a clotting activity of approximately 364% of the normal level. Base editing of FIX has also been reported. Hiramoto et al. [113] have developed a novel engineered Cas9, SpCas9-NG, which is capable of recognizing a single guanine as its protospacer adjacent motif sequence, allowing for greater flexibility in choosing the editing target. Using the engineered Cas9 nickase, the authors were then able to correct the hemophilia B-causing mutation c.947T > C (I316T), reverting the cytosine back to thymine and rescuing the FIX activity. Notably, conventional SpCas9 nickase was unable to target this mutation and produced no edited clones. Bayarsaikhan et al. [114] utilized iHLC sheets for FIX production (instead of conventional iHLC cultures), which possess increased survivability after transplantation, making them a particularly attractive model for downstream clinical translation. The FIX coding region was inserted into the APOC3 site in chromosome 11 of healthy iPSCs, which were then differentiated into iHLCs and seeded on a poly N-isopropyl acrylamide-grafted temperature-responsive cell culture dish, which promoted the formation of cell sheets upon the change in temperature from 37 °C to room temperature. The resulting cultures demonstrated superior hepatic characteristics compared to normal iHLCs, including the elevated levels of FIX secretion, 5–11 times higher than in normal iHLC cultures.

Another blood-related liver disorder is transthyretin amyloidosis (ATTR), a fatal, albeit rare, condition caused by misfolding of transthyretin. Crucially, the TTR level of expression in primary hepatocytes and iHLCs is almost identical, making the latter a particularly attractive platform for ATTR modeling. For example, using iHLCs, it was shown that ATTR onset may be influenced by the aberrant regulation of extracellular chaperone expression. One of them in particular, Serpin family A member 1 (SERPINA1), was shown by Niemietz et al. [115] to directly interact with TTR by inhibiting the conversion of TTR into insoluble high molecular forms. In a different study by the same group [116], iHLCs were used to compare the efficiency of TTR knockdown by siRNA and antisense compounds, with siRNA showing a higher efficiency of TTR silencing.

### 3.2. Monogenic Diseases with Primary Hepatic Expression and Parenchymal Damage

AATD is caused by mutations in the *SERPINA1* gene, encoding the AAT enzyme, which lead to the accumulation of the misfolded aggregated form of the protein (mutant alpha-1 antitrypsin Z, ATZ) inside the rough endoplasmic reticulum (rER) of hepatocytes, eventually resulting in their death. Importantly, AATD does not impact the viability or the hepatic differentiation potential of iPSCs, and indeed, reports exist of both fibroblasts and the Epstein–Barr virus-immortalized B lymphocytes with the AATD genotype being successfully reprogrammed into viable iPSCs [117]. It should be noted that ATZ-mediated proteotoxicity is not an omnipresent clinical manifestation of AAT, with some patients escaping this phenotype despite the presence of mutations in the *SERPINA1* gene. iHLCs generated from patient-derived iPSCs with the AATD genotype successfully recapitulate the ATZ accumulation, which simplifies the study of underlying molecular mechanisms. For example, Tafaleng et al. [118] showed that patient-specific variability in ATZ degradation efficiency exists and impacts the severity of the observed phenotype—cells where proteostatic mechanisms are robust lack the characteristic intracellular inclusions of ATZ. On the contrary, in the more severe clinical manifestations of AATD, ATZ was shown to accumulate even outside the rER. Furthermore, transcriptome and proteome comparisons of healthy and AATD iHLCs conducted by Segeritz et al. [119] showed that mitochondrial abnormalities could impact the severity of the phenotype as well. Another pathway implicated in the AATD development, as discovered by Nunzia et al. [120], is the JNK pathway. One of the previously overlooked aspects of AATD unveiled by the authors is the activation of the JNK pathway by the accumulated ATZ, which then upregulates *SERPINA1* expression through an interaction of c-Jun with a liver-specific regulatory region in the 5′-UTR (untranslated region) of the gene, creating a positive feedback loop. This was shown to occur both in a murine model and in patient-derived iHLCs, marking JNK as an additional therapeutic endpoint for the treatment of AATD. An example of iHLCs being used for pharmaceutical screenings of AATD therapies can be found in a study by Wilson et al. [121], where the response of patient-derived iHLCs to the drug carbamazepine was assessed. Carbamazepine was shown to increase the autophagic flux inside the iHLCs without impacting the actual expression level of AAT or suppressing its secretion. In addition to that, AATD iHLCs were found to be more susceptible to hepatotoxic effects of many common drugs compared to healthy iHLCs. In particular, the analgesic acetaminophen and the drugs amiodarone, danazol, puromycin, and aflatoxin-B all were significantly more toxic to AATD iHLCs than to their healthy counterparts.

Gene editing approaches have also been successfully used to repair *SERPINA1* mutations in iHLCs. For example, scarless gene editing was achieved in patient-derived iPSCs by Kaserman et al. [122] using CRISPR/Cas9 at an overall biallelic correction efficiency of 6%, producing iHLCs with reduced levels of accumulated intracellular AAT and higher levels of AAT secretion compared to their unedited counterparts. Allele-specific editing of patient-derived iPSCs is also possible, as demonstrated in a study by Smith et al. [123], wherein a specific *SERPINA1* point mutation locus (rs28929474, G > A) was successfully targeted by Cas9-gRNAs without any detectable effect on the intact wild-type allele. Another approach, demonstrated in a study by Eggenschwiler et al. [124], relied on the lentiviral-mediated delivery of AAT-targeting short hairpin RNA (shRNAs) into AATD iPSCs. The delivery of microRNA 30-styled shRNA in particular resulted in a significant and sustained reduction in AAT mRNA and protein levels, resulting in a marked reduction in the clustering of ATZ measured by fluorescence microscopy, and without significant effects on iHLCs functionality, evaluated through AAT and albumin secretion, as well as the cytochrome P450 activity. Effective bi-allelic *SERPINA1* editing in patient-derived iPSCs can also be achieved by TALENs, as demonstrated by Choi et al. [125] by employing TALEN expression vectors that specifically recognize the flanking sequences of the AATD-causing mutation, resulting in a significant reduction in AAT accumulation in iHLCs. The same study also used iHLCs for a large-scale conventional drug screening, with over 3131 total drugs tested by a high-throughput immunofluorescence analysis in a single iHLC line, out of which 380 drugs, which altered AAT level, were selected. The selection was then narrowed to 282 drugs, which decreased AAT accumulation to less than 50% of the control group, and 43 of them without side effects were selected for testing in multiple patient-derived iHLC lines. After the final screening, the five finalists that were effective in multiple cell lines were left—carbamazepine, glipizide, lithium, valproic acid, and thiamine. This impressive throughput would have been impossible to achieve with PHH cultures. Finally, Werder et al. [126] showed that adenine base editing results in significant improvement of the AATD phenotype and ER stress protection when applied both to iPSCs and to already differentiated iHLCs as well.

Alagille syndrome is caused by mutations in the components of the Notch signaling pathway, either in the ligand Jagged 1 or in the receptor Notch 2, which results in ALGS types 1 and 2, respectively. Loss of Notch signaling results in incorrect development of many organs, including the liver. Patient-derived ALGS iPSC lines have been reported [127] and can be used to study the mechanisms driving pathology in detail. For example, Guan et al. [128] used patient-derived ALGS hepatic organoids (HOs) to study the effects of *JAG1* mutations on bile duct formation, finding that ALGS mutations, most commonly via a dominant-negative effector mechanism, specifically affect the postnatal stages of liver development by disrupting the pattern of *JAG1* mRNA expression. As a result, even though mature hepatocytes can be derived from ALGS iPSCs, mature cholangiocytes and ductular structures fail to form, reflecting the similar features observed in vivo. Furthermore, following the CRISPR/Cas9 gene editing of the ALGS1 mutation, HOs regained the ability to form organized bile ducts. Sampaziotis et al. [16] managed to successfully generate cholangiocyte-like cells (CLCs) from patient-derived iPSCs in order to study the progression of the disease in the bile ducts of the liver, for which primary cultures are otherwise not readily available. The resulting CLCs were further matured through organoid formation, necessary for correct apicobasolateral polarization, and were confirmed to be representative of the native cell type through functional testing. Specifically, the MDR1-dependent transfer was confirmed by observing the accumulation of Rhodamine123, a fluorescent substrate of MDR1, in the organoid lumen; apical salt and bile transfer was demonstrated by an active export of the fluorescent bile acid cholyl-lysyl-fluorescein, while the alkaline phosphatase and gamma glutamyl transferase activities were remarkably similar to those of primary cells. Such results were previously unattainable with various differentiation protocols. Blocking the Notch signaling with the gamma-secretase inhibitor L-685 prevented the formation of organoids, demonstrating the potential of CLC organoids for ex vivo modeling of both the ALGS and the biliary system development in general.

Other biliary diseases can also be studied using iPSC-derived models. Defects in *ABCB11*, encoding the bile salt export pump (BSEP), result in elevated bile acid levels in serum due to severe intrahepatic cholestasis and may cause liver failure, referred to as progressive familial intrahepatic cholestasis type 2. Imagawa et al. [129] used patient-derived iPSCs to generate PFIC2 iHLCs, which exhibited aberrant BSEP localization and lower capacity for bile acid transfer. The authors were then able to rescue the phenotype of PFIC2 iHLCs through the addition of sodium 4-phenylbutyrate, which helps improve the membrane trafficking of the BSEP. In a somewhat more complicated study, Hayashi et al. [130] used CRISPR/Cas9 to introduce a mutation (c.3268C > T) in the R1090 codon of the *ABCB11* gene into the healthy iPSCs. The authors differentiated iPSCs into iHLCs on top of a permeable membrane of a transwell system. Much like in the previous study, the authors found the pattern of expression of the mutated BSEP to be altered—whereas it is normally mainly expressed on the apical membrane, the mutated form was localized in the cytoplasm of the iHLCs, which was later confirmed to be the case in the diseased liver tissue as well. This was shown to impact the efficiency of transcellular bile acid transfer—after TCA was added to the bottom well of the transwell system, it mostly failed to be transferred from the lower to the upper chamber after 48 h, whereas the healthy iHLCs managed to transfer the majority of it in the similar timeframe. Interestingly, TCA did not accumulate inside iHLCs during this time—rather, it was found to be exported via basolateral transfer instead, which was also confirmed to be the case for newly synthesized TCA. Taken together, these results demonstrated a novel pathophysiological mechanism of PFIC2 that could only be discovered via a relatively complex transwell culture setup.

Wilson’s disease is an autosomal recessive disorder caused by mutations in the *ATP7B* gene, encoding the ATPase copper-transporting β, which mediates copper excretion into the bile. ATP7B loss of function results in copper accumulation in the liver, which can eventually spill over into the brain. Patient-derived iPSC lines allow for comprehensive study of this disease in both the neural and hepatic tissues [131]. For example, Parisi et al. [132] used patient-derived iPSCs to study the effect of the most common *ATP7B* mutation, H1069Q, on hepatocytes. It was found that mutated ATP7B tends to localize within the Golgi apparatus of the diseased iHLCs, likely meaning that the mutation somehow supports the folding of the protein in a way that helps it to traffic into the Golgi, from where it seemingly enters the proteasome degradation. This finding suggests that the reason for the previously observed low levels of ATP7B in WD-iHLCs compared to healthy controls is the quicker turnover of the mutated ATP7B. Overeem et al. [133] used a polarized patient-derived iHLC culture, which notably forms in vivo-like canaliculi, to further clarify these underlying mechanisms of vesicular transport. It was found that the AT7B-H1069Q mutation prevents the copper-induced redistribution of ATP7B to the aforementioned bile canaliculi. Groba et al. [134] used patient-derived iHLCs to study the effects of MDR1 downregulation characteristic of WD. After the incubation of WD iHLCs with the culture medium containing 0.1 mM Cu, MDR1 expression was found to be downregulated compared to the untreated control, likely meaning that MDR1 downregulation might be an adaptive mechanism adopted by cells under high copper concentrations.

iHLCs have also been used to screen the efficiency of potential therapeutics for WD. Zhang et al. [135], for example, tested the effects of gene correction using a self-inactivating lentiviral vector and a chaperone drug, curcumin, on the iHLCs generated from patient-derived iPSCs carrying the R778L Chinese hotspot mutation in the *ATP7B*. The chosen therapy approach rescued the WD phenotype. Copper chelation treatment can also be potentially used to treat WD. However, most commercially available chelator drugs have many adverse side effects. A promising replacement in the form of methanobactin was proposed by Lichtmannegger et al. [136]. Methanobactin is a peptide with high copper affinity derived from the methanotrophic proteobacterium *Methylosinus trichosporium*, which can significantly deplete copper levels in iHLCs generated from WD patient-derived iPSCs without the typical adverse effects of commercial chelators, highlighting a potential alternative to the traditional treatment strategies.

Defects in the glucose-6-phosphate transporter lead to the development of glycogen storage disease type Ib, resulting in hypoglycemia as glucose cannot be released into circulation. In a study by Satoh et al. [137], iHLCs were generated from patient-derived iPSCs. These GSDIb iHLCs were found to faithfully recapitulate the disease phenotype, most notably the excessive intracellular accumulation of glycogen and lipids, and exhibited elevated levels of glycolytic enzymes due to the impaired glucose-6-phosphate metabolism.

### 3.3. Monogenic Diseases with Both Hepatic and Extrahepatic Expression

Niemann-Pick type C disease is a hereditary disease caused by mutations in genes *NPC1* (accounting for 95% of cases) and *NPC2*, encoding the intracellular cholesterol transport proteins, leading to misfolding and subsequent degradation by the ER-associated degradation machinery and the accumulation of cholesterol in lysosomes and endosomes. Soga et al. [138] produced patient-derived iPSC lines and generated iHLCs from them. NPC-iHLCs demonstrated lower ATP levels, reduced mitochondrial membrane potential as measured by the JC-1/Mitotracker CMXRos staining ratio, and impaired autophagy, assessed using the p62 and LC3 expression measurements. The cells were then used to screen the effectiveness of different 2-hydroxypropyl-cyclodextrins for cholesterol accumulation treatment. It was demonstrated that 2-hydroxypropyl-α-cyclodextrin had no effect on cholesterol accumulation, while hydroxypropyl-β-cyclodextrin (HPBCD) and 2-hydroxypropyl-γ-cyclodextrin (HPGCD) were equally effective at reducing cholesterol accumulation at high (>100 uM) concentrations. Interestingly, despite the similar efficiency, HPGCD and HPBCD treatments resulted in differing, albeit largely similar, patterns of gene expression. Völkner et al. [139] used patient-derived iPSCs with the homozygous I1061T mutation to screen the effectiveness of various pharmacological chaperones, which the cells with this mutation are particularly responsive to, in iHLCs. Both iHLCs with the I1061T mutation and the heterozygous E612D/P543Rfs*20 and Y394H/Y394H displayed reduced expression level and retention of the mutated peptide in the ER, which were successfully ameliorated by PC treatment. Finally, Maetzel et al. [140] used TALEN-mediated genetic correction to repair the same functional I1061T mutation in patient-derived iPSCs. The differentiated iHLCs demonstrated a rescue of the NP-C phenotype and showed normal cholesterol distribution and a correct response to serum treatment, resulting in a suppression of SREBP2 cleavage. In addition, autophagy induction by starvation, mTOR inhibition, and mTOR-independent autophagy enhancers (such as carbamazepine, verapamil, trehalose, and SMER28) was also tested and was all found to enhance p62 clearance in all cases.

Mitochondrial DNA depletion syndrome is caused by mutations in multiple genes. Guo et al. [141] used patient-derived iPSCs to investigate the effect of mutations in the gene encoding the deoxyribonucleoside kinase (DGUOK). Deficiency in this protein results in iron overload, which progresses to liver failure via as yet unclear mechanisms. Healthy, diseased, and gene-corrected hepatic organoids were generated. All types of organoids exhibited similar mitochondrial morphology; however, diseased ones had significantly reduced mtDNA copy number and expression level, as well as reduced membrane potential and lower ATP production, confirming that disease phenotype is primarily influenced not by mitochondrial quality, but rather by mtDNA copy number. The exact mechanism of iron overload-related cell death was established to be connected to NCOA4-dependent degradation of ferritin in lysosomes, exacerbated by the depletion of glutathione in diseased iHLC organoids, which increases lipid peroxidation and results in an increased sensitivity to ferroptosis. Jing et al. [142] tried to improve hepatocyte function in patient-derived DGUOK-deficient iHLCs by NAD treatment. NAD reproducibly increased the expression of electron transport chain genes and the ATP levels, rescuing the mitochondrial dysfunction and reducing ROS levels in the process. The therapeutic mechanism was found to be the modulation of peroxisome proliferator-activated receptor gamma coactivator 1-alpha (PGC1α) activity by NAD, which promotes PGC1α activation through the deacetylation of sirtuin 1. In turn, PGC1α activates a number of transcription factors important for mitochondrial function, such as the estrogen-related receptor alpha, peroxisome proliferator-activated receptor alpha, and nuclear respiratory factors, both activating them by binding and increasing their expression levels via a feedback loop mechanism. Another similar mitochondrial disorder with a pronounced impact on the liver is the Alpers-Huttenlocher syndrome, caused by mutations in the mitochondrial DNA polymerase gamma, responsible for the replication of mtDNA. In a study by Li et al. [143], patient-derived iPSCs were used to investigate the seemingly paradoxical mechanism of AHS suddenly becoming completely untreatable and typically resulting in liver failure upon valproic acid (VPA) exposure, normally used as an antiepileptic drug. The authors found the level of active cleaved caspase-9 (and its substrate, caspase-3) to be much higher in AHS iHLCs after the VPA treatment, indicating the mitochondrial nature of the involved apoptosis pathway. Indeed, in addition to other impairments of mitochondrial physiology, like the levels of OPA-1 complex and mtDNA being less abundant in AHS iHLCs, mPTP was transiently activated much more commonly in AHS iHLCs than in a control group, triggering bursts of superoxide generation and increasing the apoptotic sensitivity of cells. Interestingly, inhibition of mPTP also reduced the percentage of apoptotic cells after VPA treatment, suggesting it to be causative of the observed hepatotoxicity.

Cystic fibrosis is caused by mutations in the cystic fibrosis transmembrane regulator (CFTR), resulting in increased bile viscosity and the occlusion of ducts, which in the case of the liver is particularly evident in bile ducts. Therefore, CLCs are the most relevant model of iPSC-derived liver CF. Ogawa et al. [144] used patient-derived iPSCs carrying the most common CF-causing ΔF508 mutation to generate functional CLCs, which were found to form cysts in an impaired manner, typical of CF. Even when cyst formation was induced by forskolin, many cysts were not hollow and contained branching structures. The authors then tested the experimental CF drug VX809 (small molecule), which was primarily designed to work in lungs, and confirmed that while it did not affect the aberrant cyst formation, it did rescue the reduced levels of CFTR protein in cells. In a previously mentioned study by Sampaziotis et al. [16], iPSC-derived CLC organoids with the same mutation, ΔF508, were used to model CF. Despite the confirmed transcription of the *CFTR* gene, very low levels of CFTR protein were detected in CF organoids, confirming the rapid ER degradation of the CFTR protein characteristic of the disease. This study also confirmed the effectiveness of VX809 for CFTR restoration in CF CLCs. Finally, Fiorotto et al. [145] also used patient-derived iPSCs with the ΔF508 mutation for the screening of CF treatments. In addition to VX809 treatment, which resulted in moderate improvements in CF phenotype, a similar small molecule drug, VX770, was tested with comparable results. Furthermore, the authors discovered the Src activity in CF CLCs to be elevated, leading to elevated NF-kB activity, which resulted in the delocalization of the actin cytoskeleton and the increased secretion of MCP-1 and IL-8, chemoattractants for pro-inflammatory immune cells. Src inhibition with the PP2 was able to restore the actin cytoskeleton, which as a side effect made treatment with small molecules more effective as well.

Pompe disease, or glycogen storage disease type II, is a disorder caused by mutations in the *GAA* gene, encoding the lysosomal acid α-glucosidase (GAA). Deficiency in GAA leads to the accumulation of glycogen in the cytoplasm and lysosomes. Patient-derived iPSC lines for this disease are known to exist [146] and can be used to study it ex vivo.

Ouchi et al. [147] used patient-derived liver organoids to model the effects of FGF19 treatment on Wolman disease, caused by defects in lysosomal acid lipase, which leads to lethal steatohepatitis and liver fibrosis. Wolman disease organoids, consistent with the clinical phenotype, rapidly accumulated lipids. This was shown to be treatable either by administering the recombinant LAL protein or, preventively, through FGF19. Interestingly, the magnesium treatment did not have any notable positive effects on the progression of the disease. Abetalipoproteinemia, caused by mutations in the Microsomal Triglyceride Transfer Protein (MTTP), similarly results in excessive lipid accumulation in hepatocytes and cardiomyocytes. Liu et al. [148] used patient-derived iHLCs to model this disease, finding that they correctly recapitulate the phenotype of ABL, characterized by undetectable MTTP activity, absence of both intracellular and secreted apoB, reduced hepatic lipid secretion, and excessive lipid accumulation. Correction of the C136G mutation by CRISPR/Cas9 gene editing mostly normalized these changes.

Primary hyperoxaluria type 1 is caused by a deficiency in the alanine glyoxylate aminotransferase (AGT), a liver-specific enzyme, resulting in excessive accumulation of insoluble oxalate products in the kidney and the urinary tract. Estève et al. [149] used patient-derived iPSCs carrying a c.731 T > C mutation (p.I244T) in exon 7 of the *AGXT* gene to test the efficiency of gene therapeutic approaches in treating PH1. The mutated form AGT in PH1 iHLCs was found to be unstable, quickly forming aggregates that were then degraded, contributing to the low protein levels of the functional enzyme. The addition of the lentiviral vector carrying the replacement *AGXT* gene to the cells was able to rescue the expression of AGT.

### 3.4. Viral and Other Liver Diseases

The modeling of viral liver diseases is distinct from the described studies due to the external nature of the pathogen, meaning that healthy iPSCs are used for the modeling of infectious diseases.

Liver hepatitis infections are widely studied using iPSC models. Shlomai et al. [150] directly compared the functionality of iHLCs and PHHs in that context, finding that both were permissive to Hepatitis B virus (HBV) infection and, crucially, possessed an innate type I immune response, something that hepatoma cell lines, such as HepG2, notably lack. However, it should be noted that the magnitude of the innate immune response is somewhat lacking in iHLCs, suggesting that the maturation of hepatic function also correlates with that of the innate immune axis. Nonetheless, iHLCs are a promising model for studying both the life cycle of HBV and the potential treatment strategies, and they also display less individual variability compared to PHHs. For example, Chen et al. [151] used iHLCs to study the mechanisms of viral entry of the HBV into liver cells. The authors confirmed that EGF signaling modulates the process of viral entry via the sodium taurocholate co-transporting polypeptide (NTCP) by enhancing the HBV attachment to cells in a dose-independent manner, with epidermal growth factor receptor (EGFR) acting as a co-receptor to NTCP, and determined the exact endocytosis pathway involved in the viral internalization to be the same as for the EGFR endocytosis. Specifically, using pathway-specific inhibitors, it was shown that depending on the EGF concentration, either clathrin-mediated (low dose of EGF) or clathrin-independent (high dose of EGF) pathways can occur, with the CIE in particular being followed by lysosomal degradation of both EGFR and the HBV particles.

An important aspect of viral disease modeling is the ability of the chosen cell line to support long-term infection and viral replication. Primary hepatocytes can only do so for 5–10 days, whereas iHLCs, as demonstrated by Xia et al. [152], maintain this ability to some extent for up to 30 days, with 20–24 days being the mark where the efficiency of infection starts declining. For even longer-term studies, such as the one conducted by Nie et al. [153], iPSC-derived hepatic organoids may be preferable. These organoids show infection susceptibility comparable to that of primary hepatocytes and exhibit similar dynamics of virus release, and, crucially, possess enhanced maintenance of hepatic function in organoids compared to iHLCs and especially primary cultures, making them a good long-term infection model that does not dedifferentiate. In contrast to iHLCs, induced pluripotent stem cell-derived hepatic progenitor cells (iHPCs) are relatively resilient to HBV infection owing to their low NTCP expression. Overexpression of NTCP, as suggested in a study by Kaneko et al. [154], can make them more susceptible to infection, providing another way to study long-term viral dynamics in liver cells, as iHPCs maintain their hepatic phenotype for much longer than iHLCs. Furthermore, iHPCs also possess a functional IFNα innate immune response against the HBV. The authors used these hepatic progenitor cell (HPC) properties to investigate the long-term persistence of HBV covalently closed circular DNA (cccDNA) in host cells, the retention of which is thought to be linked with the resistance of HBV to antiviral treatments. After 5 passages (and 52 days of culture), cccDNA still persisted in host cells via an unclear mechanism, suggesting that HPC models can recapitulate this property of the HBV infection. It was also demonstrated that cccDNA levels were directly impacted by the JAK signaling pathway, which affects the aforementioned innate immune activity, with the cccDNA levels being significantly higher in HPC cultures where it was inhibited.

iHLCs can be used for antiviral drug screenings. Sakurai et al. [17] used iHLCs to assess the effectiveness of HBV infection suppression by entecavir and Myrcludex-B, both known anti-HBV drugs. The addition of both drugs to the iHLC cultures resulted in reduced HBV levels in culture media in the case of entecavir, which inhibits reverse transcription, and a decrease in viral mRNA levels in the case of Myrcludex-B, which blocks HBV entry into the cells, without any apparent hepatotoxicity in either case.

Wu et al. [155] showed that iHLCs are also permissive to an infection by the Hepatitis C virus (HCV). The infection of iHLCs was found to be persistent and could be supported for up to 21 days. The exact conditions during the hepatic differentiation that enable the cells to become permissive for HCV infection were identified in this study as the activation of miR122, EGFR/EphA2, and PI4KIIIα and downregulation of IFITM1. Sangiamsuntorn et al. [156] characterized the molecular profile of iHLCs used for HCV infection modeling. The authors demonstrated that the main differences enabling the support of HCV infection for longer than by their PHH counterparts were the elevated, stable expression of both the functional hepatocyte markers (namely the CYP isoforms, OATP2, and MRP2) and the HCV receptors (claudin-1, occludin, TAPA-1 (CD81), scavenger receptor B1, ApoB, and ApoE). This allowed the HCV not only to infect iHLCs but also to complete its life cycle and produce progeny. The innate immune response is also helpful for accurate HCV modeling. Sakurai et al. [157] elaborated on the exact mechanisms of the anti-HCV IFN immune response observed in iHLCs, identifying type III IFNs to be the most upregulated compared to type I IFNs. Schwartz et al. [158] also determined that HCV infection induced an antiviral inflammatory response, finding that infected iHLCs persistently secrete TNF-α. Notably, the expression of *IL-28B* was found to be elevated in iHLCs during the 2-week period following the infection, before eventually declining. Schöbel et al. [18] used iHLCs to study HCV-host interactions in the context of innate immunity and lipoprotein synthesis. HCV infection was found to activate the IFN immune response, the intensity of which could be reduced by JAK inhibition by counteracting the induction of interferon-stimulated genes. Infected iHLCs were also found to secrete lipoprotein particles, reminiscent of VLDL, and exhibit a similarly high triglyceride composition. In a study by Yoshida et al. [159], the viability of iHLCs as a platform for anti-HCV therapy screening was assessed. The authors found that anti-CD81 antibodies and IFN were effective for attenuating the infection, while IgG treatment was not.

Hepatitis E viral (HEV) infection can also be recapitulated using iHLCs. In a study by Helsen et al. [160], iHLCs were confirmed to support the entirety of the HEV replication cycle, which resulted in an elevated IFN innate immune response. The HEV replication in iHLCs was confirmed by the treatment with ribavirin, a replication inhibitor, which markedly reduced the number of both intra- and extracellular HEV RNA copies. Dao Thi et al. [161] used an iHLC culture system to demonstrate that sofosbuvir, a drug used to treat HCV infection, is also functional for HEV treatment, particularly when combined with ribavirin, despite it being less effective than for its intended purpose of HCV treatment. Wu et al. [162] compared the infection permissiveness of iHLCs for multiple HEV genotypes, successfully infecting both with the primary isolates from infected animals and the cell culture-adapted Kernow-C1 P1 supernatant from HepG2 cells. Interestingly, the replication of these primary HEV isolates was not restricted by the cyclophilin A, which otherwise inhibited the replication of P1-HEV when inactivated by cyclosporine A, showing that the source of a viral isolate may have a pronounced effect on the results of an ex vivo experiment.

iHLCs can also be used to model other viral infections unrelated to hepatitis. For example, Yang et al. [163] used iPSC-derived cells to assess the permissiveness of various cell types to SARS-CoV-2 in a large screening study, which included the hepatocyte and cholangiocyte organoids among the utilized models. The authors demonstrated high levels of viral sgRNA in both hepatocyte and cholangiocyte organoids, suggesting a robust SARS-CoV-2 infection and proliferation. Infected hepatocytes demonstrated a pronounced induction of multiple chemokines with upregulation of IL-17, TNF, and NF-κB signaling, as well as a downregulation of key hepatocyte markers and metabolic pathways. Infected cholangiocytes exhibited a similar expression pattern, with upregulated cytokine levels, inflammatory pathways, and IL-17 signaling.

In summary, iPSCs are a unique platform for investigating viral diseases, facilitating the dynamic evaluation of dose-dependent viral load effects, pertinent not only to viral liver diseases, like hepatitis, but also to various other viral infections affecting liver cells at heightened risk. Finally, iPSCs can be used for the modeling of diseases that do not necessarily fit into the previously defined categories but are still worth highlighting. For example, since fetal cholangiocytes are involved in the formation of cystic liver lesions, characteristic of polycystic liver disease, iPSC-derived CLC organoids with an appropriate genotype can also be used to screen the efficiency of PLD treatments, which are primarily focused on the reduction in cysts. Sampaziotis et al. [16] treated CLC organoids with secretin, somatostatin, and octreotide, a synthetic analog of somatostatin used in clinical treatment of PLD. As expected, secretin, which is known to increase the secretory activity of cholangiocytes, resulted in the increase in organoid size, while somatostatin and octreotide, which, conversely, reduce the cholangiocyte secretory activity, decreased the size of organoids instead, correctly reproducing the effects of the drug ex vivo.

Tetralogy of Fallot is a congenital heart disease caused by mutations in the *JAG1* gene and is known to be associated with ALGS, with 7–13% of ALGS patients also developing TOF. Guan et al. [79] studied this interaction, finding that a common TOF-causing Gly274Asp *JAG1* mutation has no effect on the liver organoid formation, despite the mutated protein being unable to activate the Notch pathway due to an altered glycosylation pattern, while iPSCs carrying the C829X and ALGS2 *JAG1* mutations were unable to form liver organoids, highlighting the variability in clinical manifestations of different *JAG1* mutations.

Finally, in a study by Tian et al. [164], a biliary atresia model was created both from patient-derived and from healthy iPSCs with *GPC1* and *ADD3* genes knocked out. Biliary epithelial cells generated from these iPSCs exhibited reduced capacity for biliary tissue formation, producing reduced ductal structure and forming no detectable tubes in a 3D differentiation assay, as well as exhibiting increased fibrogenesis and lower expression of biliary markers, such as CK7, CK19, and epithelial cell adhesion molecule. Furthermore, the expression of YAP1 was found to be elevated in these cells, and, interestingly, the protein was shown to be colocalized with collagen 1, which is normally absent from healthy biliary cells altogether. The authors were able to attenuate the levels of both the YAP1 and the collagen 1 by treating the cells with an anti-fibrogenic drug, pentoxiphylline. These findings are particularly important given the previous lack of well-reported human BA models and the inability of rodent models to correctly recapitulate the severity of BA-associated liver fibrosis.

## 4. In Vivo Studies

Animal models play a vital role in studying the pathophysiology of diseases and testing new drugs under physiological conditions. Rodents are most often used for these purposes, but studies are also conducted on zebrafish [165], pigs [166], and macaques [167]. There are several approaches to obtaining animal models: induction of disease using hepatotoxic compounds [168,169] and various diets [170,171], surgical approaches [172], creation of transgenic [173,174,175] and chimeric animals [176,177], and combinations of various approaches [178]. It should be noted that currently available in vivo models cannot fully reproduce all aspects of the human disease [59,179], in connection with which the development of ex vivo modeling continues. iPSCs are actively used in preclinical trials for the study and therapy of a number of diseases, including liver diseases. Basically, researchers create mutant animal lines or induce diseases by trauma or exposure to toxins, after which they treat the disease by transplanting healthy iPSC-derived cells. Figure 3 shows the strategies for treating monogenic and viral liver diseases in vivo based on iPSCs as described in the published studies. Disease-specific iPSC-derived cells are also used to model pathologies by transplanting these cells into healthy mice [180]. In turn, to assess the engraftment and expression of target proteins in vivo, healthy human cells are used, which are transplanted into healthy mice [181,182]. It is worth noting that in all studies that included transplantation, with the exception of Wu et al. [183], the authors transplanted the cells once (without subsequent transplants), after which they studied the cell engraftment and expression of factors (coagulation factors, human albumin) and conducted functional tests (tail clip challenge).

The in vivo section is divided into three parts: studies devoted to hemophilia, studies in which cells were transplanted into sick non-hemophilic animals, and studies in which cells were transplanted into healthy animals. In all in vivo studies devoted to hemophilia (with the exception of the study by Lyu et al. [181]), the authors transplanted cells into animals with hemophilia. Most of the studies involving cell transplantation in animals (Table 3) are devoted to the study of hemophilia (12 studies), of which 7 studies are devoted to hemophilia A and 5 studies to hemophilia B. This may be due to the fact that human factors FVIII and FIX successfully replace the corresponding mouse factors in hemophilic mice [184,185].

It is known that a large fraction of factor VIII in blood is produced by liver sinusoidal endothelial cells [185]. This is probably why almost all studies (with the exception of Qiu et al. [186]), where induced pluripotent stem cell-derived mesenchymal stem cells (iMSCs) were transplanted for hemophilia A, included transplantation of corrected iECs. Various functional tests are used to assess the effectiveness of iPSC-based therapy. One of the most useful tests for assessing FVIII/FIX activity in mice is the tail bleeding test (tail clip challenge). Authors Kim et al. [187] transplanted 1 million human iECs differentiated from patient iPSCs and edited patient iPSCs into the tail vein of FVIII-deficient immunodeficient mice and then measured factor VIII activity using a chromogenic assay and tail clip challenge. A single transplant of edited iECs resulted in amelioration of the hemophilia phenotype for more than 3 months. Park et al. [188] inverted the FVIII mutant in patient iPSCs and then transplanted patient-derived corrected iECs by subcutaneous injection into the hind limb of B6 HA mice. Three out of nine mice that received the edited cells survived the tail-clip challenge (endpoint—2 days); the remaining six also showed increased survival (on average 111 min) compared with control mice that did not receive any cells or received cells derived from non-corrected patient iPSCs. Adult and neonatal HA mouse line B6 (null) were injected subcutaneously with 2–3 million HA-corrected-iECs, respectively, in the study of Rose et al. [189]. The cells remained integrated into the host vascular system for at least 10–16 weeks and expressed FVIII, the GFP reporter, and CD31 (endothelial marker). The titer of coagulation factor VIII remained high (11.2–369.2% of the standard—healthy human plasma) at 2 weeks post-transplantation, and treated mice showed reduced blood loss in a tail-clip bleeding test compared with non-transplanted HA mice. To restore FVIII function, Son et al. [190] used, in addition to edited, patient-derived iPSCs or 3D liver organoids.

Following subcutaneous injection of 2–4 million iECs or the corresponding number of 3D liver organoids (2888) into the thigh of immunocompetent HA mice, mice showed good cell survival (>3 months) and reversed bleeding phenotypes against lethal wounding challenges. In another study, Olgasi et al. [191] transplanted 2 million patient-derived corrected iECs into the portal vein of NOD-SCID (non-obese diabetic severe combined immunodeficient) null HA mice, after which activated partial thromboplastin time (aPTT, a blood test that characterizes coagulation of the blood) was measured at 3, 6, 9, and 12 weeks after transplantation. The mice showed a stable increase in factor VIII activity, which remained stable for up to 12 weeks post-transplantation. Restoration of factor VIII activity was also demonstrated after intraperitoneal injection of iECs combined with microcarrier beads supporting their viability.

In addition to editing iPSCs, Talmon et al. [192] used another strategy: first, HA patient-derived mononuclear cells were edited using lentiviral delivery, and only then were the cells reprogrammed and later differentiated into iECs. Transplanted HA-corrected Flk-1-GFP+ iECs in NOD-SCID HA mice were detected in the liver for up to 1 month post-transplantation. Due to the insufficient number of studies in which the authors first edited the cells and then reprogrammed them for subsequent transplantation, it is difficult to assess how much more optimal this strategy is than editing after reprogramming. One way or another, iECs are widely used for the treatment of hemophilia A-positive mice.

Unlike other authors who used iECs, Qiu et al. [186] transplanted intravenously 2 million of CM-Dil (dye used for cell tracking) labeled iMSCs derived from FVIII-corrected iPSCs and BDDF8-iPSCs to B6 FVIII knockout mice. They also performed a tail-bleeding assay, collected plasma 1, 2, 3, and 4 weeks after transplantation, and determined aPTT. Unlike other studies, the authors isolated internal organs (heart, liver, spleen, lung, and kidney) from the treated and HA mice. CM-Dil-positive cells were detected by immunofluorescence analysis in the liver and lung sections of mice treated with iMSCs 1, 2, and 3 weeks after transplantation. Long-term engraftment of corrected iMSC with restoration of FVIII function and phenotypic rescue was observed in HA mice that received the cells.

**Table 3 ijms-26-09432-t003:** Some applied aspects and results of in vivo studies.

Disease	Mouse Model	Cell Type	Cell Number, mln	Injection Site	Assessment Type, Duration, Result	Reference
HA	FVIII −/− ID mice	iECs	1	IV injection in tail vein	Factor VIII activity was measured using a chromogenic assay and tail clip challenge. A single transplant of edited iECs resulted in amelioration of the hemophilia phenotype for more than 3 months	[187]
	FVIII −/− IC mice	iECs, 3D liver organoids	2–4	SC injection	Good cell engraftment (more than 3 months) and reversed bleeding phenotypes against lethal wounding challenges	[190]
	B6 FVIII −/− mice	CM-Dil labeled iMSCs	2	IV injection	Performed a tail-bleeding assay, collected plasma 1, 2, 3, and 4 weeks after transplantation, and determined aPTT. Long-term engraftment of corrected iMSC with restoration of FVIII function and phenotypic rescue was observed in HA mice transplanted with cells	[186]
	Adult and neonatal B6 FVIII −/− mice	iECs	10; 2–3	SC injection	The titer of coagulation factor VIII remained high (11.2–369.2% of the standard—healthy human plasma) at 2 weeks post-transplantation, and treated mice showed reduced blood loss in a tail-clip bleeding test compared with non-transplanted HA mice	[189]
	FVIII −/− NOD/SCID mice	iECs	2	IV injection in the peritoneal cavity	Mice showed a stable increase in factor VIII activity, which remained stable up to 12 weeks after transplantation (aPTT test). Recovery of factor VIII activity was also demonstrated after intraperitoneal injection of iECs combined with microcarrier beads supporting their viability	[191]
	B6 FVIII −/− mice	iECs	N/A	SC injection	Three of the nine mice that received edited cells survived the tail-clip challenge (endpoint—2 days); the remaining six also showed increased survival (on average 111 min) compared to control mice that did not receive any cells or received cells derived from non-corrected patient iPSCs	[188]
	FVIII −/− NOD/SCID mice	iECs	N/A	N/A	HA-corrected iECs were detected in the liver up to 1 m post-transplantation	[192]
HB	B6 FIX −/− mice	iHLCs	0.1	Transcutaneous injection into the liver	ELISA and chromogenic assay were performed on plasma of mice sacrificed 2 weeks after transplantation. As a result, restoration of FIX activity in vivo was demonstrated	[111]
	FIX −/− ID (RGFKO) mice	iHLCs	N/A	IS injection	A chromogenic test was used to show the bleeding phenotype being rescued; the clotting FIX concentration was 0.7 μg/mL (nearly equal to a non-HB mouse)	[184]
	B6 FIX −/− mice	iHLCs	2	Injection under the kidney capsule	The clotting activities of transplanted mice were approximately 5% of the wild-type values (2 weeks post transplantation), and the bleeding time was shorter than that of non-transplanted mice.	[193]
	NOD/SCID mice	iHLCs	42 per kg	IS injection	Cells not only successfully engraft but also secrete hFIX. Only two weeks after transplantation the hFIX antigen was detected, and after four weeks or more, it was no longer detectable in any group	[181]
	FIX −/− mice	iHLCs	0.5	IS injection	Two sequential IS injections with a 1-week interval of 5 × 10^5^ of healthy murine iHLCs to HB mice. The recipient mice were sacrificed at 1 to 4 weeks after transplantation. Hemostatic function was assayed by thromboelastography using the citrated kaolin mode. The authors studied a set of parameters: the time to initial fibrin formation, the speed to reach a specific level of clot strength, and others. iHLCs transplantation improved hemostasis measures, thrombus formation, and coagulation factor IX activity. The transferred cells were located in the liver of recipient HB mice	[183]
HCV	ID MUP-uPA/SCID/Bg mice	iHLCs	4	IS injection	The authors infected mice with HCV after IS injection of healthy iHLCs, creating an in vivo model for studying hepatitis, and measured HCV RNA titers and HCV core antigen concentrations in the sera of mice and showed that it is human iHLCs that carry the virus, since mice cannot be infected with HCV.	[194]
CTLN1	ID (NSG) mice	Hepatic organoids	2	SC injection	The authors demonstrated successful engraftment using immunostaining of GFP and human-specific albumin antibodies separately	[93]
ASA	NOD/SCID/IL2Rγ− mice	iECs	0.4	SC injection	As expected, ASLD iECs showed decreased ability to form blood capillaries and arterioles in vivo	[180]
GSDII	ID (NSG) mice	iMPs	0.5	IM injection	Myogenic progenitors engrafted into murine muscle formed human myofibers	[195]
Crigler-Najjar syndrome	Gunn rats with mutated Ugt1a1	iMSCs	4	IS injection	The iMSCs engrafted and survived in the liver for up to 2 months. The expression of several human-specific hepatocyte markers, including albumin, demonstrated that the transplanted iMSCs differentiated into functional hepatocytes	[196]
AD	Arg1Δ mice	Murine iHLCs	2	IS injection	Even though iPSCs with Arg1Δ alleles in their genome were successfully repaired, mice that received iHLC showed an insignificant recovery in urea cycle function when compared to control mice, and some mice’ survival in this lethal model was prolonged by only up to a week	[197]
AATD	ID (Alb-uPA+/+; Rag2−/−; Il2rg−/−) mice	iHLCs	0.5	IS injection	Successful engraftment was demonstrated by human albumin being detected in the serum of transplanted animals for at least 5 weeks. Authors demonstrated the potential of combining human iPSCs with genetic correction to generate clinically relevant cells for autologous cell-based therapies	[198]
-	ID (NSG) mice	IH-ICC organoids	0.5 per 1 scaffold	IH injection	The organoids formed vascularized tissue following intrahepatic injection into ID NSG mice	[199]

AATD—alpha-1 antitrypsin deficiency, AD—arginase deficiency, aPTT—activated partial thromboplastin time, ASA—argininosuccinate lyase deficiency, CTLN1—citrullinemia type 1, ELISA—enzyme—linked immunosorbent assay, GSDII—glycogen storage disease type II, HA—hemophilia A, HB—hemophilia B, HCV—hepatitis C virus, IC—immunocompetent, ID—immunodeficient, iECs—iPSC-derived endothelial cells, IH—intrahepatic, IH ICC—iPSC-derived hepatic progenitor—inverted colloid crystal, iHLCs—iPSC-derived hepatocyte-like cells, IM—intramuscular, IS—intrasplenic, IV—intravenous, iMSCs—iPSC-derived mesenchymal stem cells, NOD SCID—non-obese diabetic severe combined immunodeficient, NSG—NOD SCID gamma, RGFKO—RAG2−/−IL2Rγ-/YFAH−/−, and SC—subcutaneous.

Various routes of administration of iPSC-based drugs have been used for the treatment of hemophilia B. Intrasplenic injection of patient-derived, gene-corrected iPSC-HLCs was performed on Rag2−/− IL2rγ−/− Fah−/− FIX−/− hemophilic mice by Ramaswamy et al. [184]. The authors demonstrated these iPSC-HLCs to be viable and functional in mouse models for an impressive period of 9–12 months. Not only was the FIX activity characterized, but also the presence of hAlb (a surrogate marker for the efficiency of engraftment) by enzyme-linked immunosorbent assay (ELISA) of mouse plasma and immunohistochemistry of mouse liver, respectively. The chromogenic test was used to show the rescue of the bleeding phenotype; the clotting FIX concentration was 0.7 μg/mL (nearly equal to non-HB mouse). For testing hiPSCs, not only immunodeficient animals are used, but also induced immunosuppression with cyclosporine A is used [193]. Okamoto et al. [193] used C57BL6 FIX-KO mice, which were transplanted with 2 × 10^6^ of healthy human iPSC-derived HLCs under the kidney capsule. The authors demonstrated that the clotting activities of transplanted mice were about 5% of the values of C57BL6 wild-type mice (2 weeks post transplantation), and the bleeding time was shorter than that of non-transplanted mice. Gene-corrected iPSC-HLCs were transplanted not only into diseased mice but also into healthy mice to assess the success of engraftment and hFIX expression in vivo. Lyu et al. [181] sequenced the genome of iPSCs corrected by CRISPR-Cas9 using WGS to ensure that no unwanted mutations were created by editing. A total of 4.2 × 10^7^/kg of patient-derived iHLCs were transplanted to non-hemophilic NOD/SCID mice through splenic injection to show that cells not only successfully engraft but also secrete hFIX and to evaluate the presence of human Alb in the liver tissue. Only two weeks after transplantation the hFIX antigen was detected, and after four weeks or more, it was no longer detectable in any group. Luce et al. [111] studied the activity of the coagulation factor IX in both monolayer culture and organoids of non-corrected and corrected iHLCs. 10^5^ of corrected hepatocytes were then transplanted by transcutaneous injection into the liver of HB B6 mice, and 1U of recombinant FIX was added to ensure the survival of hemophilic mice. ELISA and chromogenic assay were performed on plasma of mice sacrificed 2 weeks after transplantation. As a result, using a chromogenic test, restoration of FIX clotting activity in mouse plasma to the level of healthy mice (about 0.7 μg/mL) was shown. Unlike HA, hemophilia B is treated by transplantation of iPSC-derived HLCs.

Although in most discussed studies only one injection of cells was performed, Wu et al. [183] made two sequential intrasplenic injections one week apart of 5 × 10^5^ of healthy mouse iPSC-derived hepatocytes to HB mice. The animal model was preconditioned using immunosuppression (monocrotaline and rifampicin); the recipient mice were sacrificed at 1 to 4 weeks post-transplantation. Hemostatic function was assayed by thromboelastography using the citrated kaolin mode. The authors studied a set of parameters: the time to initial fibrin formation, the speed to reach a specific level of clot strength, and others. iHLCs transplantation improved hemostasis measures, thrombus formation, and coagulation factor IX activity. The transplanted cells were located in the liver of recipient HB mice.

Another 9 studies involved cell transplantation into animals in the context of other inherited liver diseases (slightly more often without parenchymal involvement): argininosuccinate lyase deficiency, Pompe disease, Crigler-Najjar syndrome, arginase-1 deficiency, alpha-1 antitrypsin deficiency, hepatitis C, and 1 study without diseases.

Below are the studies, the authors of which transplanted cells into diseased animals. Wal et al. [195] successfully injected 20 μL of 5 × 10^5^ of patient (Pompe disease) and gene-corrected iPSCs-derived myogenic progenitors into the tibialis anterior muscle of NOD scid gamma (NSG) immunodeficient recipient mice that had been pre-injured with BaCl_2_. The expression of human lamin A/C, human spectrin, or human dystrophin (human-specific nuclear and sarcolemmar antigens) was evaluated by immunohistochemistry. It was demonstrated that myogenic progenitors engraft into murine muscle to form human myofibers. Intrasplenic injection of 2 million gene-edited (TALEN) mouse iPSC-derived HLCs restored urea cycle function in arginase-1-deficient mice (tamoxifen-inducible Arg1-Cre mice) [197]. Even though iPSCs with Arg1Δ alleles in their genome were successfully repaired, mice that received iHLCs showed only a slight recovery in urea cycle function when compared to control mice, and some mice’s survival in this lethal model was prolonged by only up to a week. To assess the liver regeneration of immunocompetent female Gunn rats with mutated Ugt1a1 (phenotypically close to Crigler–Najjar syndrome) after intrasplenic injection of 4 million human iMSCs, Spitzhorn et al. [196] analyzed samples by ELISA specific for human ALB and many other methods. The iMSCs engrafted and survived in the liver for up to 2 months. The expression of several human-specific hepatocyte markers, including albumin, hepatocyte nuclear factor 4α, and others, demonstrated that the transplanted iMSCs differentiated into functional hepatocytes. These results suggest that human iMSC transplants may aid in liver regeneration in vivo, making them a potentially useful therapy option for hereditary liver disorders. Thus, in mice, human iPSC-derived cells (including MSCs and myogenic progenitors) can be successfully used to treat genetic diseases other than hemophilia.

The studies described below involve transplanting healthy cells into healthy animals. Some studies involve in vivo disease modeling or transplanting edited cells into immunodeficient animals to assess the success of cell engraftment. Yusa et al. [198] were among the first (2011) to study the transplantation of human iPSC-derived hepatic cells and the regeneration of damaged liver. The authors used the example of Alb-uPA+/+;Rag2−/−;Il2rg−/− mice injected with 5 × 10^5^ patient-derived (AATD) gene-corrected iPSC-derived HLCs. Successful engraftment was demonstrated by human albumin detection in the serum of transplanted animals for at least 5 weeks. The authors demonstrated the potential of combining hiPSCs with genetic correction to generate clinically relevant cells for autologous cell-based therapies. iPSCs are also used to model diseases in vivo. Four million hiPSC-derived HLCs were injected into the spleens of MUP-uPA/SCID/Bg mice by Carpentier et al. [194]. The authors showed that the corrected iPSCs could efficiently differentiate to hepatocyte-like cells and engraft into the animal model without tumor formation. The authors infected mice with HCV after intrasplenic injection of healthy iHLCs, creating an in vivo model for studying hepatitis C. Carpentier et al. measured HCV RNA titers and HCV core antigen concentrations in the sera of mice and showed that it is human iHLCs that carry the virus, since mice cannot be infected with HCV. Kho et al. [180] created plugs containing both Matrigel and healthy or ASLD iECs plugs and transplanted them (200 mcl of Matrigel solution containing nearly 0.4 × 10^6^ cells) into the lower abdominal region of immunodeficient (NOD/SCID/IL2Rγ−) mice. Authors counted the number of loops with positive CD31 staining using images of the entire plug. As expected, ASLD iECs showed decreased ability to form blood capillaries and arterioles in vivo.

In a number of similar studies, the authors transplanted liver organoids into animals, which not only took root successfully but also participated in vascularization (formation of blood vessels). Ng et al. [199] created iPSC-derived hepatic progenitor-inverted colloid crystal organoids and showed that the organoids were able to be infected by HCV and formed vascularized tissue following intrahepatic injection to immunodeficient NSG mice, but the infected organoids were not transplanted. In the work of Akbari et al. [93] devoted to the study of citrullinemia type 1 ex vivo, the authors were able to partially restore the healthy phenotype of patient-derived eHEPOs using lentiviral delivery of ASS1 ex vivo. The authors transplanted healthy cells into healthy mice: immunodeficient NSG mice were transplanted with 2 million eHEPO (organoid) cells, obtained from healthy iPSCs by subcutaneous injection. The authors demonstrated successful engraftment using immunostaining of GFP and human-specific albumin antibodies separately.

We have demonstrated that iPSC-derived cells are widely used to model and treat hereditary liver diseases in vivo. Human cells successfully engraft and express proteins of interest, and as a result, phenotype improvement is observed. This suggests that a transition from preclinical studies to clinical studies of hereditary liver disease treatment will soon begin.

## 5. Discussion

In our review, we examined the results of iPSC-based studies on liver disease modeling and therapy; however, other types of stem cells, embryonic stem cells (ESCs) and MSCs, can also be used for these purposes. iPSCs have advantages over ESCs: not only can the autologous cells be obtained, but they are also more ethically acceptable (do not require the destruction of the embryo). In comparison with another type of stem cell, MSCs, iPSCs do not require complex invasive procedures to be obtained [200]. Unlike MSCs and ESCs, iPSCs can be used to create a disease model of a specific patient (personalized diagnostics and medicine).

Due to their pluripotency, iPSCs and ESCs can differentiate into all types of somatic cells, including liver cells. For example, both iPSCs and ESCs have been used in a number of studies to obtain HLCs [155,182,194,201], cholangiocytes [144], or MSCs to support liver function restoration [196]. Disadvantages of iPSCs compared to ESCs sometimes include the retention of certain methylation patterns (the so-called epigenetic memory) and residual gene expression from the somatic cells from which they were obtained [202,203]. These can theoretically be retained during differentiation and lead to unwanted changes in the properties of differentiated cells (e.g., functional activity). At the same time, the preservation of epigenetic patterns of parent cells in iPSCs does not apparently contribute to the emergence of unique characteristics of iPSC-derived somatic cells in comparison to endogenous cells and cells differentiated from ESCs [204]. Differentiated cells from isogenic iPSCs and somatic cell nuclear transfer-derived embryonic stem cells were shown to display comparable differentiation efficiency, gene-expression heterogeneity, physiological properties, and metabolic functions [204]. iPSCs have a similar methylation pattern to ESCs (but different from the parent cells), and iPSCs derived from each of the three developmental germ layer tissues differentiate into functionally similar cells [182]. And according to Lin et al. [203], iPSC-derived germline cells recapitulate epigenetic programming and gene expression patterns corresponding to equivalent endogenous germ cell types.

MSCs have a more limited potency than iPSCs but can also successfully differentiate into HLCs [205] and ECs [206]. Currently, there are no approved drugs based on iPSCs, ESCs, and MSCs for the treatment of liver diseases. However, in the context of the treatment of liver diseases in vivo, the possibility of transplantation of both MSCs (Spitzhorn et al. [196] (Crigler-Najjar syndrome), Qiu et al. [186] (Hemophilia A)) (with the expectation of in vivo differentiation) and cells differentiated from MSCs [200] has been shown. Interestingly, one of the most promising strategies is to obtain MSCs from iPSCs for subsequent transplantation into mice with genetic disorders (Crigler-Najjar syndrome).

Among the concerning potential disadvantages of iPSCs that hinder the emergence of approved cell therapy drugs based on iPSCs are their notorious potential for tumorigenicity [207], which iPSC-derived somatic cells may also have [208,209], and immunogenicity, a characteristic not only of allogeneic (associated with HLA donor–recipient mismatch) but also of autologous cells [210].

Possible transformation of iPSCs into tumor cells may be caused by the presence of genomic integrations [211] (in the case of integrative reprogramming methods) and genomic rearrangements as a result of reprogramming. Exogenous Yamanaka factors are silenced in both iPSCs and differentiated cells but can be expressed in residual partially reprogrammed pre-iPSCs [212]. The probability of tumor formation can be reduced by using non-integrative reprogramming methods or selective media. For example, since human hepatocytes are able to synthesize glucose and arginine from galactose and ornithine, a special medium containing galactose and ornithine can be used to purify the final product from undifferentiated iPSCs [213]. Another approach to increasing the safety of iPSC use is the alternative combination of transcription factors used for reprogramming. iPSCs obtained by delivering three Yamanaka factors, Oct-4, Klf4, and Sox-2 (without c-Myc), successfully differentiate into iHLCs and do not cause tumorigenesis post-transplantation into mice [214]. In order to reduce the tumorigenic potential of iPSC-derived somatic cells, it is necessary to perform whole-genome sequencing of the obtained cells for genomic abnormalities, use only cells that do not express surface markers of pluripotency (sorting), and conduct preliminary experiments on animals.

The immunogenicity of iPSCs may be caused by the presence of epigenetic abnormalities, genomic rearrangements, and somatic mutations leading to the expression of aberrant antigens [215]. To prevent the immunogenicity of iPSCs, in addition to modulating the patient’s immune system, it is possible to make changes to the cells themselves, for example, inactivation of the genes of the major histocompatibility complex [216].

iPSCs are used not only for clinical studies as potential cell therapy but also for drug screening and preclinical studies. Differentiation of iPSCs into iHLCs is used in most of the reviewed studies in the context of liver disease modeling. iPSC-derived hepatocyte-like cells have a number of advantages over primary and immortalized hepatocytes. Among the main benefits of iHLCs are their high similarity with primary hepatocytes (albeit somewhat limited by their fetal phenotype) [84] and availability of reproducible cell production protocols [217,218,219]. iHLCs can be used both ex vivo and in vivo on their own [14,193,194,220] or as 3D cell cultures [93,128,147,199]. Within our classification of liver diseases, iHLCs are most frequently used to study and treat monogenic diseases with primary hepatic expression and viral diseases. For example, Hayashi et al. [130] demonstrated impaired BSEP compartmentalization and TCA transport in patient-derived (PFIC) iHLCs and normalization of these processes after *ABCB11* gene editing, while Schöbel et al. [18] showed that iHLCs can be used as a physiologically relevant model to study HCV-host interactions.

iHLCs are less commonly used to study and treat certain disease types. This may be due to the fact that iHLCs are immature, which limits their ability to fully recapitulate all the functions of mature hepatocytes [221]. For example, iHLCs were shown to exhibit lower albumin production, incomplete urea cycle activity, and lower cytochrome P450 enzyme activity, which is critical for the use of iHLCs in drug screening [222]. These indicators can probably be improved by changing the activity of a number of signaling pathways that influence them [223,224]. For instance, progesterone induces CYP2A6, CYP2B6, CYP2C8, CYP3A4, and CYP3A5 expression in PHHs [225]. At the same time, it is known that progesterone receptor expression occurs during the early stages of reprogramming and is further regulated during the process of differentiation [226].

There are differences in differentiation efficiency of different iPSC lines [227], which may hinder the study of certain diseases. Monogenic diseases with both hepatic and extrahepatic expression and other diseases include damage not only to the liver parenchyma but also to other parts of the liver (e.g., bile ducts) or other organs. These diseases are less frequently modeled using iPSCs in general and iHLC in particular; differentiation of iPSCs into other cell types is more common: for example, cholangiocytes [18,144].

In addition, the role of hepatocyte interactions with the cellular environment in the pathogenesis of these diseases is potentially higher than in other groups; therefore, the use of organoids is more effective for modeling [141,147]. Organoids partially imitate the physiological features and microarchitecture of native tissues [90], which opens up great potential for modeling liver diseases. The preservation of tissue architecture in HOs can be important in drug discovery as well, which is especially evident when chemometric imaging methods are used. For example, LaLone et al. [228] used quantitative Raman imaging to assess spatially resolved biomolecule spectra in PHHs and HOs, finding significant differences between them, both under physiological conditions and in response to the addition of different drugs. Furthermore, HOs can also be used to study parasitic infections. For example, Yang et al. [229] used hepatocyte-derived HOs to study the liver stages of the *Plasmodium falciparum* life cycle and the parasite-hepatocyte interactions. Normal PHH cultures cannot be sustained long enough for the parasite to progress to the mature schizont stage, something that was shown to be possible in HOs, enabling both the study of transcriptome changes in response to infection and the screening of the antiparasitic drugs to be conducted using this model. Significant improvements in liver organoid maturity have recently been achieved by depositing organoids on the pillar plate platforms. Shrestha et al. [230] used these platforms in conjunction with microarray 3D bioprinting to achieve up to a 50-fold reduction in cell culture volume required to achieve HO formation compared to the conventional Matrigel-based protocols, with the resulting organoids exhibiting superior viability, albumin expression, and CYP3A4 activity. At the same time, organoid formation is more complex and expensive than iHLC production, so in general, the percentage of studies where hepatic organoids obtained from iPSCs are applied in modeling and treatment of hepatic disease remains small.

Most animal studies utilizing cell therapy also use iHLC, although it has been shown in the example of hepatocyte injection into the portal vein and splenic artery that this can lead to portal hypertension and the possibility of gastric and splenic infarction [231]. Organoids are used for transplantation much less frequently, probably due to the length and complexity of the collection process and the need for complex surgical manipulations [199] or strictly controlled administration conditions associated with the risk of embolism [232].

Currently, preclinical studies involve experiments on animals (in vivo). Animal models are more relevant than stem cell-based models and allow for functional tests to be conducted and evaluated, but their use is often associated with a number of difficulties. Despite the overall greater physiological complexity of animal models even compared to advanced culture systems, some aspects of human disease nonetheless cannot be accurately represented in laboratory animals due to the significant cross-species differences [233]. Overall, roughly 29% of human genes and 25% of murine genes expressed in the liver lack a definite cross-species homolog. For example, while 36 pathways overlap between human and mouse livers during the progression of non-alcoholic fatty liver disease, none are enriched in both simultaneously, with regulation ranging from concordant enrichment, or lack thereof, or even opposite regulation [234].

One such aspect is the differences in hepatocyte metabolism and biochemical functions. In our review we have previously highlighted these discrepancies in ammonia metabolism and in the urea cycle, but other differences also exist. For example, cytochrome isoform expression was shown to vary considerably between animal models and humans. A comparative study by Hammer et al. [235] showed that CYP2B6 and CYP3A4 expression is much higher in PHHs than in laboratory animal models. Crucially, differences were also found in CYP response to pharmacological treatment. CYP2E1 was found to be downregulated in mice (but not in rats or HepaRG cells) in response to cyproconazole treatment via an unknown mechanism, while prochloraz was observed to exhibit a greater impact on human CYP1A1 compared to the rat CYP1A1, measured by the fold change. Albadry et al. [236] additionally found differences in CYP enzyme expression patterns and histological localization between humans and murine model animals, concluding that while CYP2E1 was relatively conserved between humans, mice, and rats, CYP1A, CYP2C, CYP2D, and CYP3A showed considerable differences in cross-species expression patterns. Therefore, while liver spatial metabolic compartmentalization in general seems broadly comparable between rodents and humans, as reviewed by Cunningham and Porat-Shilom [237], subtle and often poorly characterized (partially due to the limited access to human tissue compared to the abundance of animal studies) functional differences do exist. Hence, careful consideration when choosing a model for a study of a particular liver disease, especially when pharmacological aspects are concerned, is crucial. At the same time, the widespread use of iHLCs in vivo in the context of the treatment of hemophilia A and B may be associated with the similarity in the structure of the FVIII and FIX proteins of humans and mice: 70 and 75%, respectively (Protein Data Bank, Pairwise Structure Alignment), and their functional cross-reactivity.

Parenchymal damage and fibrosis can arise as a result of multiple etiologies, and not all mechanisms are transferable between animal models and humans, as reviewed by Lee et al. [238]. In particular, in addition to the aforementioned differences in CYP expression and activity, higher sensitivity of human hepatic cells to fibrotic factors and lipopolysaccharide, as well as differences in gut microbiome, also contribute to the formation of a gap between murine and human pathophysiology of liver fibrosis, one that may be bridged through the use of iPSC-derived 3D liver cultures. Screenings of drug hepatotoxicity in particular are also seen as benefitting from being conducted in 3D iPSC-derived liver models [239].

Viral-host interactions are especially difficult to model in animals, since some human viruses do not infect cells of other species. For example, no immunologically tractable animal model of HCV infection currently exists, with the closest equivalent being either the non-primate *Hepacivirus* naturally circulating in horses (which is not by any means a readily available or an easily tractable model) or various engineered mice models, which require a blockade of most host immune responses in order to function [240]. Even for those human viruses where infection can occur in a mouse model, like HBV, the response to infection is markedly different, with no spread of infection occurring and only mild hepatitis being present, as reviewed by Du et al. [241]. This leaves human cell cultures as the only reasonable option for the study of these diseases.

Many of these discrepancies can be partially overcome by the use of humanized mice [242,243]; however, the throughput of this approach is even lower compared to the use of regular mice. As a result, it is not as useful for early screening of drug candidates and gene therapeutic approaches, something that iHLCs excel at.

In the context of future development of iPSC-based approaches to modeling and treating liver diseases, it is worth mentioning relatively new methods of hepatic tissue engineering, including several types of bioprinting and liver-on-a-chip cluster of technologies. Various 3D bioprinting techniques are used to create in vitro liver models: Taymour et al. used coaxial extrusion-based 3D bioprinting to create a HepG2-based liver sinusoid-like model [244], Skardal et al. created a protocol for extrusion bioprinting of a hyaluronic acid and gelatin-based hydrogel system containing different types of hepatic cells [245]. Wang et al. 3D-bioprinted iHLCs within two types of scaffolds (polymeric and liver-derived) [246]. A significant contribution to the creation of artificial hepatic tissues is made by the selection of materials that create an imitation of the cellular microenvironment and the basis for the entire structure (tissue scaffolds): biomolecules (gelatin, collagen, cellulose, etc.), synthetic polymers (polyvinyl alcohol, poly-N-isopropylacrylamide, poly-L-lactic acids, etc.), and decellularized extracellular matrix derived from liver [247].

Three-dimensional bioprinting techniques are also used to create organ-on-chips, a promising technology that has been developing rapidly over the past two decades. There are many varieties of liver-on-chips [248], including liver chip based on matrixless 3D spheroid culture [249,250], liver chip based on layer-by-layer deposition [251,252], etc. Kogler et al. [253] created “organ-in-a-column” by printing HOs mixed with glass beads directly into liquid chromatography column housings, enabling the literal on-column online monitoring of cellular metabolic response to added compounds, such as heroin. Among the disadvantages of 3D hepatic models, including liver-on-chips, it is worth noting the difficulty of standardization and the associated regulatory restrictions for use as therapeutic drugs [254]. Other disadvantages of liver-on-chips varieties include not entirely accurate metabolic zonation in the microfluid flow or nonspecific absorption of drugs being tested [255]. However, further development of these technologies, especially with the use of modern biomaterials, can not only reduce the existing disadvantages but also create more advanced organ models and body-on-chip/human-on-chip [256].

To summarize, the iPSCs platform, with its many advantages and some disadvantages, is already successfully used to model and treat certain liver disorders. This promising, fast-paced field of biomedicine is helping understand the pathology underlying liver disease, test new therapies, cure liver disorders, help long-suffering patients, and eventually alleviate the burden on modern healthcare.

## Figures and Tables

**Figure 1 ijms-26-09432-f001:**
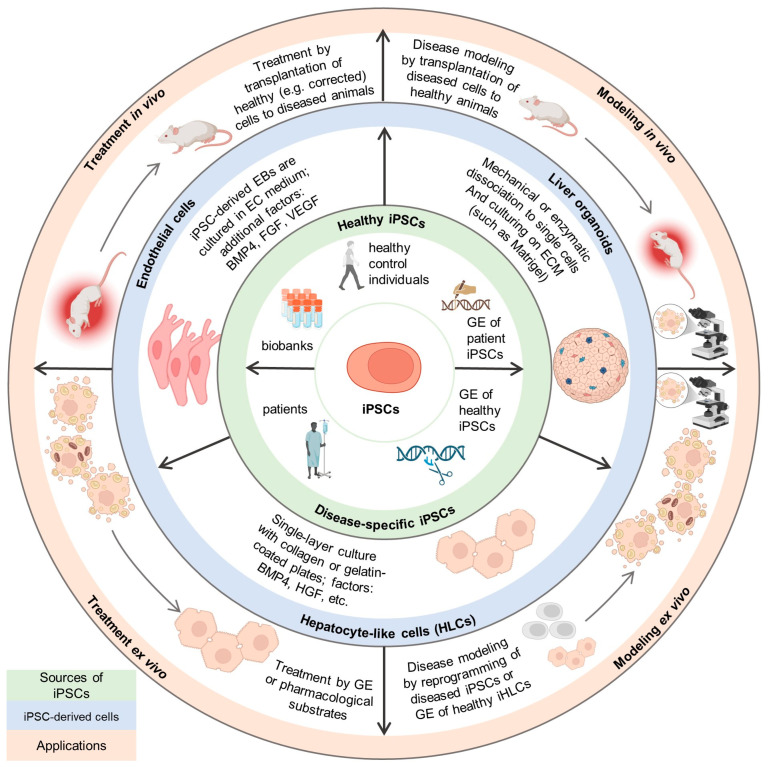
iPSCs in modeling and treatment of liver diseases. In this context, iPSCs can be conditionally divided into two groups: healthy and disease-specific. Healthy iPSCs are obtained from biobanks, by reprogramming healthy individuals’ cells or genetic engineering (GE) of patient iPSCs, whereas disease-specific iPSCs are obtained by reprogramming patient cells or GE of healthy iPSCs. iPSCs may be differentiated in a wide range of 2D cell cultures and 3D organoids, including iHLCs, liver organoids, and endothelial cells. Four application directions based on the iPSC platform can be defined: modeling ex vivo, treatment ex vivo, modeling in vivo, and treatment in vivo.

**Figure 2 ijms-26-09432-f002:**
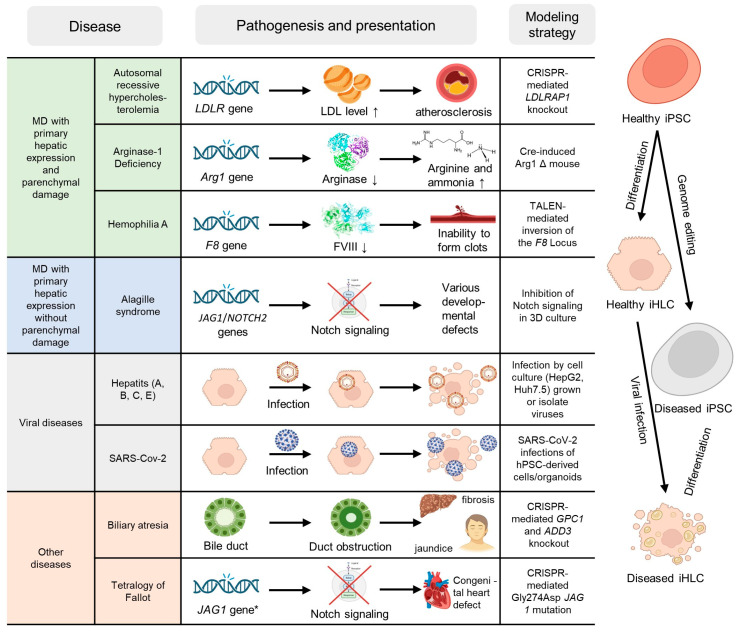
Strategies for ex vivo modeling of various liver diseases based on iPSCs (excluding reprogramming of patient cells). Pathogenesis and presentation are depicted schematically. MD—monogenic diseases, TALEN—transcription activator-like effector nucleases, CRISPR—clustered regularly interspaced short palindromic repeats, LDL—low-density lipoprotein. * One of the gene variants, mutation in which leads to the development of Tetralogy of Fallot [91].

**Figure 3 ijms-26-09432-f003:**
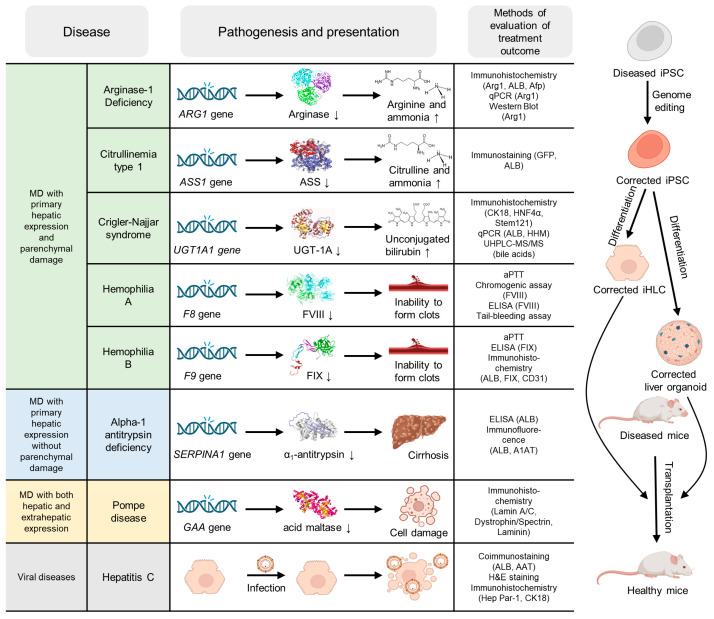
In vivo treatment strategies for various types of liver diseases based on iPSCs. The pathogenesis and presentation of diseases are schematically depicted, and methods for assessing the outcome of in vivo treatment are presented. MD—monogenic diseases, qPCR—quantitative polymerase chain reaction, UHPLC—MS/MS—ultra high-performance liquid chromatography coupled with mass spectroscopy/ mass spectroscopy, aPTT—activated partial thromboplastin time, ELISA—enzyme-linked immunosorbent assay.

**Table 1 ijms-26-09432-t001:** Various types of hereditary, viral, and other liver diseases are discussed in this review.

Disease Group	Disease	Disease Prevalence	Phenotype	Studies Discussed in This Review		
				Ex Vivo		In Vivo
				Modeling	Treatment	
Monogenic diseases with primary hepatic expression without significant parenchymal damage	Hemophilia A	1:5000 (mostly males) [30]	Deficiency of coagulation factor VIII leads to impaired blood clotting and increased bleeding	+	+	+
	Hemophilia B	1:20,000 (mostly males) [31]	Factor IX deficiency causes prolonged bleeding after injuries, surgery, or even spontaneously	+	N/A	+
	UCDs	1:35,000 including ASL deficiency) [32]	Mutations lead to deficiencies of the various enzymes and transporters involved in the urea cycle, causing hyperammonemia or the buildup of cycle intermediates	+	+	+
	Hypercholesterolemia	1:250 (mostly familial heterozygous) [33]	High levels of blood cholesterol as a result of autosomal recessive and familial hypercholesterolemia	+	+	N/A
	PH1	1–3:1,000,000 [34]	Increased secretion of oxalate caused by mutations in the *AGXT* gene leads to kidney stones and potential kidney damage	N/A	+	N/A
	WTTA	1:4 (very aged patients) [35]	Misfolded TTR deposits in various tissues mostly affect the heart and tendons of the elderly adults	+	N/A	N/A
	FAP	10,186 patients (extrapolated globally) [36]	Also known as TTR-related amyloidosis. Buildup of abnormal amyloid deposits in the nervous system and other organs causes pain and muscular weakness and may affect the kidneys and the heart	N/A	+	N/A
	Crigler-Najjar syndrome	1:1,000,000 [37]	Improper processing of bilirubin leads to an increase in bilirubin in the blood, causing potential brain damage and severe jaundice	+	N/A	+
	GSDIa	1:125,000 [38]	Lack of release of glucose during fasting and accumulation of excess glycogen and fat in the liver and kidney provoke severe hypoglycemia and other metabolic pathologies	+	N/A	N/A
Monogenic diseases with primary hepatic expression and parenchymal damage	AATD	2–5:10,000 [39]	Deficiency in alpha-1 antitrypsin results in lung and liver diseases such as COPD and cirrhosis	+	+	+
	ALGS	20–33:1,000,000 [40]	Lack or complete absence of bile ducts is accompanied by accumulation of bile in the liver, which leads to severe heart and/or liver disease	+	+	N/A
	PFIC	1–2:100,000 [41]	Impaired bile acid transport and secretion from the liver associated with inadequate bile accumulation and liver disease	+	+	N/A
	ASA	1:70,000 [42]	Urea cycle disorder caused by deficiency or absence of the enzyme ASL may provoke progressive liver damage	N/A	+	+
	WD	1:30,000 [43]	Copper is accumulated due to its defective transport to bile and excretion in waste products, causing symptoms related to the brain and liver	+	+	N/A
	GSDIb	1:500,000 [38]	A deficiency in the G6PT results in disrupted glucose homeostasis and immunological impairment, particularly characterized by neutropenia and neutrophil dysfunction	+	N/A	N/A
	TYR1	5–6:600,000 [44,45]	Inability to effectively break down the amino acid tyrosine. Accumulation of tyrosine and its byproducts leads to severe health consequences, including liver and renal disorders	+	N/A	N/A
Monogenic diseases with both hepatic and extrahepatic expression	MDDS, AHS	1–2:10,000 [46]	Mutations in the *POLG* gene, crucial for mitochondrial DNA replication and repair, lead to progressive developmental regression, uncontrollable seizures, and liver degradation, typically occurring in infants and young children.	N/A	+	N/A
	NP-C	4–6:600,000 [47]	The accumulation of cholesterol and glycolipids caused by mutations in *NPC1* and *NPC2* impairs the body’s ability to process and transport them. This buildup mostly impacts the brain, liver, and spleen, resulting in a variety of systemic and neurological symptoms	N/A	+	N/A
	Wolman disease	1:500,000 [48]	Deficiency in LAL provokes the accumulation of cholesteryl esters and triglycerides in various tissues and organs. Gastrointestinal symptoms and liver and spleen enlargement are present	N/A	+	N/A
	Pompe disease	1:18,711 [49]	Also known as GSDII. Deficiency in the acid GAA causes a buildup of glycogen in cells, leading to muscle and nerve cell damage in the body	+	N/A	+
	CF	1–2:6000 [50]	Mutations in both alleles of the gene encoding the CFTR protein lead to the impaired mucus clearance from the lungs and the colonization of the lungs by bacteria. The pancreas, liver, kidneys, and intestine are also affected	N/A	+	N/A
	Abetalipoproteinemia	1:1,000,000 [51]	Mutations in the microsomal *MTTP* are associated with the impaired absorption of fats and fat-soluble vitamins, causing low or absent levels of plasma cholesterol, LDLs, and VLDLs. The symptoms affect the gastrointestinal system, nervous system, and eyes.	N/A	+	N/A
Viral hepatitis	Hepatitis B	254 million people worldwide (2022) [52]	Viral infections that cause liver inflammation. Hepatitis viruses B and C cause chronic disease, whereas hepatitis E is usually self-limiting and resolves within 2–6 weeks	N/A	+	N/A
	Hepatitis C	50 million people globally [53]		+	+	+
	Hepatitis E	19.47 million cases of acute hepatitis E (AHE) globally in 2021 [54]		N/A	+	N/A
Other disorders	BA	5–10:100,000 [55]	The bile ducts of a child are narrowed, blocked, or absent, causing liver failure or cirrhosis	+	N/A	N/A
	TOF	3:10,000 [56]	Congenital heart anomaly with four specific cardiac defects	N/A	+	N/A
	PLD	1–10:1,000,000 [57]	Numerous cysts are present in liver tissue, which may cause sudden pain, inflammation, and other symptoms	N/A	+	N/A

The “+” sign means that this review discusses works devoted to modeling or treatment of the disease ex vivo/in vivo. N/A means that we are not aware of such studies or they are not available. UCDs—urea cycle disorders, PH1—primary hyperoxaluria type 1, WTTA—wild-type transthyretin amyloid, FAP—familial amyloid polyneuropathy, GSDIa—glycogen storage disease type Ia, GSDIb—glycogen storage disease type Ib, GSDII—glycogen storage disease type II, AATD—alpha-1-antitrypsin deficiency, ALGS—Alagille syndrome, PFIC—progressive familial intrahepatic cholestasis, ASA—argininosuccinate lyase deficiency, WD—Wilson’s disease, AHS—Alpers-Huttenlocher syndrome, NP-C—Niemann–Pick type C, CF—cystic fibrosis, CFTR—cystic fibrosis transmembrane conductance regulator, BA—biliary atresia, PLD—polycystic liver disease, COPD—chronic obstructive pulmonary disease, TTR—transthyretin, TYR1—tyrosinemia type 1, MDDS—mitochondrial DNA depletion syndrome, TOF—tetralogy of Fallot, ASL—argininosuccinate lyase, G6PT—glucose-6-phosphate transporter, LAL—lysosomal acid lipase, GAA—acid alpha-glucosidase, LDL—low-density lipoprotein, and VLDL—very low-density lipoprotein.

**Table 2 ijms-26-09432-t002:** Comparison of different 2D ex vivo liver models.

Cell Type	Relevance	Ease of Acquisition and Use	Throughput	Loss of Viability in Long-Term Culture
Primary hepatocytes [66,67,68,69,70]	High, but cells dedifferentiate rapidly in a matter of days	Difficult to acquire and handle	Very low	Occurs quickly
Immortalized hepatocytes [71,72,73,74,75]	Comparatively low, especially under 2D conditions, cells are highly variable	Inexpensive, easy to acquire and handle	Very high	With adequate technique, culture lifespan is unlimited
iPSC-derived hepatocyte-like cells [76,77,78]	Adequate; cells are somewhat variable and tend to exhibit a fetal phenotype	Expensive and moderately difficult to acquire and handle	Medium-low	Occurs, but is comparatively slower

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
