# Peer review of "Application of Induced Pluripotent Stem Cells (iPSCs) in Hereditary and Viral Diseases of the Liver: Modeling and Treatment"

_ijms, 2025, doi:10.3390/ijms26199432_

Round 1

Reviewer 1 Report

Comments and Suggestions for Authors

In the manuscript titled "Application of induced pluripotent stem cells (iPSCs) in hereditary and viral diseases of the liver: modeling and treatment," the authors explored the use of iPSCs as a versatile platform for modeling and treating a wide range of inherited and viral liver diseases.

Overall, the manuscript is a comprehensive and well-documented review, demonstrating a deep understanding of the scientific literature on the application of iPSCs in hepatology. Its main strength is the in-depth analysis of individual studies, providing specific details on methodologies, mutations, and key findings for a wide variety of pathologies.

However, some issues should be addressed to enhance the manuscript's impact and clarity.

Abstract

The abstract needs to be more incisive. The authors should focus not only on what they did (i.e., "review") but, more importantly, on what they found and why their analysis is significant. They should also state that despite the success for some pathologies, challenges remain for other viral diseases, such as chronic hepatitis B, and for the vast majority of hereditary diseases where cell therapy could make a difference.

Justification of Scope

The justification that "Previously published reviews... often focused on a limited number of liver diseases or discussed a limited number of studies" is insufficient. The authors must specify the gaps that their review fills and clearly outline the unique contribution of their work.

Summary Table

To avoid a dense, heavy text, the information on all the studies cited in sections could be summarized in tables. The columns could include: Disease, iPSC Type (patient/healthy), Model (2D/3D), Objective (Modeling/Treatment), Method (Drug/Gene), Key Finding, and Reference. This would provide a quick and organized overview. This section is an excellent example of exhaustive literature research but needs to be synthesized and reorganized to improve readability and impact.

Integrating Broader Perspectives

The manuscript could also be improved by discussing its findings in a broader and more complete light, integrating cutting-edge research to provide a more comprehensive perspective. This would be a significant value-add. The review highlights the importance of 3D organoids for overcoming the limitations of 2D cultures, particularly for achieving liver maturity. The reference DOI: 10.1007/s12015-024-10776-6 suggests that progesterone signaling may be a key factor not only for reprogramming but also for subsequent differentiation. Progesterone signaling can influence developmental pathways and the formation of complex tissues, resulting in more mature and functional liver organoids, with increased enzymatic activity (e.g., cytochrome P450) or improved structural organization, thus overcoming one of the main limitations of iHLCs, as repeatedly highlighted in the review.

Author Response

Dear Reviewer 1:

Thank you for allowing us to submit a revised version of our manuscript titled “Application of induced pluripotent stem cells (iPSCs) in hereditary and viral diseases of the liver: modeling and treatment”. We appreciate the time and effort you put into reviewing our manuscript as well as the valuable comments, which will improve its quality. Following your comments, we made various changes, which are now highlighted for your convenience and also explained in our point-by-point reply below.

Sincerely,

Vasiliy Reshetnikov

Comment 1: Abstract The abstract needs to be more incisive. The authors should focus not only on what they did (i.e., "review") but, more importantly, on what they found and why their analysis is significant. They should also state that despite the success for some pathologies, challenges remain for other viral diseases, such as chronic hepatitis B, and for the vast majority of hereditary diseases where cell therapy could make a difference.

Reply:  Thank you for pointing this out. We are presenting a comprehensive analysis of 2D and 3D iPSC-based products in the context of liver diseases, discussing the advantages and disadvantages of this platform, including the comparison with other types of stem cells and animal models. The abstract was corrected.

Comment 2: Justification of Scope The justification that "Previously published reviews... often focused on a limited number of liver diseases or discussed a limited number of studies" is insufficient. The authors must specify the gaps that their review fills and clearly outline the unique contribution of their work.

Reply:  Indeed, the Introduction section did not fully disclose the relevance and novelty of our study. This section has been supplemented with information regarding the authors' contribution to the field of modeling and treatment of liver diseases using iPSCs (page 2, lines 71-83)

Comment 3: Summary Table To avoid a dense, heavy text, the information on all the studies cited in sections could be summarized in tables. The columns could include: Disease, iPSC Type (patient/healthy), Model (2D/3D), Objective (Modeling/Treatment), Method (Drug/Gene), Key Finding, and Reference. This would provide a quick and organized overview. This section is an excellent example of exhaustive literature research but needs to be synthesized and reorganized to improve readability and impact.

Reply:  Thank you for this comment. We agree that the detailed description of ex vivo and in vivo studies should be duplicated in tables. Our manuscript includes a table for the in vivo block (Table 3) presented in the main manuscript and supplementary Table S1 with the detailed description of all analyzed ex vivo studies.

Comment 4: Integrating Broader Perspectives The manuscript could also be improved by discussing its findings in a broader and more complete light, integrating cutting-edge research to provide a more comprehensive perspective. This would be a significant value-add. The review highlights the importance of 3D organoids for overcoming the limitations of 2D cultures, particularly for achieving liver maturity.

Reply: Thank you for this valuable comment that we completely agree with. We expanded the discussion section and included the study by LaLone et al. (use of Raman spectroscopy of PHHs and hepatic organoids), Yang et al. (hepatic organoids can be used for studying parasitic infections), and Shrestha et al. (using pillar plate platforms for improvement of liver organoid maturity) (pages 30-31, lines 1146 – 1165). We also added the study by Kogler et al., in which the authors printed hepatic organoids into a chromatographic column for online monitoring of cellular metabolic response to added compounds (pages 32, lines 1248 – 1251).

Comment 5:

The reference DOI: 10.1007/s12015-024-10776-6 suggests that progesterone signaling may be a key factor not only for reprogramming but also for subsequent differentiation. Progesterone signaling can influence developmental pathways and the formation of complex tissues, resulting in more mature and functional liver organoids, with increased enzymatic activity (e.g., cytochrome P450) or improved structural organization, thus overcoming one of the main limitations of iHLCs, as repeatedly highlighted in the review.

Reply: Thank you for this comment, the results of this study were used in the Discussion section (pages 30, lines 1130 – 1135).

Reviewer 2 Report

Comments and Suggestions for Authors

This comprehensive and well-structured review provides an overview of the current state of iPSC technology in the context of liver disease. The authors have covered a vast array of hereditary and viral liver disorders, as well as modeling strategies. They have discussed therapeutic interventions, including gene editing and pharmacological approaches. The review is highly relevant, considering the growing interest in personalized medicine. Tables and figures have helped to simplify complex information. The review is well-referenced with recent publications. My comments are:

The resolution of figures needs to be high enough for publication.

Section 4 (In vivo studies): The text sometimes mixes descriptions of studies where cells were transplanted into diseased vs. healthy animals. This should be revised. Also, an introductory paragraph dividing the in vivo studies into these two categories would improve clarity.

While the manuscript deeply analyzes iPSC-based clinical trials, it does not adequately discuss the limitations to clinical application, including safety concerns, tumorigenicity, and immunogenicity.

Author Response

Dear Reviewer 2:

Thank you for allowing us to submit a revised version of our manuscript titled “Application of induced pluripotent stem cells (iPSCs) in hereditary and viral diseases of the liver: modeling and treatment”. We appreciate the time and effort you put into reviewing our manuscript as well as the valuable comments, which will improve its quality. Following your comments, we made various changes, which are now highlighted for your convenience and also explained in our point-by-point reply below.

Sincerely,

Vasiliy Reshetnikov

Comment 1: The resolution of figures needs to be high enough for publication.

Reply:  All of the Figures (in TIFF format) in the manuscript meet the IJMS’ requirements of resolution 300dpi.

Comment 2: Section 4 (In vivo studies): The text sometimes mixes descriptions of studies where cells were transplanted into diseased vs. healthy animals. This should be revised. Also, an introductory paragraph dividing the in vivo studies into these two categories would improve clarity.

Reply: We completely agree with your comment. Indeed, the review lacked clarity while discussing transplantation into diseased vs. healthy animals. We made all the necessary changes in section 4 In vivo studies (pages 23, lines 857-861).

Comment 3: While the manuscript deeply analyzes iPSC-based clinical trials, it does not adequately discuss the limitations to clinical application, including safety concerns, tumorigenicity, and immunogenicity.

Reply: Thank you for this comment. We agree that our review did not sufficiently discuss the factors that hinder the application of iPSCs in clinical practice. The Discussion section was updated to include a discussion of the factors hindering the implementation of iPSCs into clinical practice (tumorigenicity and immunogenicity) and possible ways to overcome these limitations (pages 29-30, lines 1086-1111).

Round 2

Reviewer 1 Report

Comments and Suggestions for Authors

N/A